# Foxp3 depends on Ikaros for control of regulatory T cell gene expression and function

Rajan M Thomas[1], Matthew C Pahl[1], Liqing Wang[2], Struan FA Grant[1,3], Wayne W Hancock[2], Andrew D Wells[1,2]*

[1]Center for Spatial and Functional Genomics, The Children's Hospital of Philadelphia, Philadelphia, United States; [2]Department of Pathology, Perelman School of Medicine at the University of Pennsylvania and The Children's Hospital of Philadelphia, Philadelphia, United States; [3]Department of Pediatrics, Perelman School of Medicine at the University of Pennsylvania and The Children's Hospital of Philadelphia, Philadelphia, United States

*For correspondence:
adwells@pennmedicine.upenn.edu

**Competing interest:** The authors declare that no competing interests exist.

**Abstract** Ikaros is a transcriptional factor required for conventional T cell development, differentiation, and anergy. While the related factors Helios and Eos have defined roles in regulatory T cells (Treg), a role for Ikaros has not been established. To determine the function of Ikaros in the Treg lineage, we generated mice with Treg-specific deletion of the Ikaros gene (*Ikzf1*). We find that Ikaros cooperates with Foxp3 to establish a major portion of the Treg epigenome and transcriptome. Ikaros-deficient Treg exhibit Th1-like gene expression with abnormal production of IL-2, IFNg, TNFa, and factors involved in Wnt and Notch signaling. While *Ikzf1*-Treg-cko mice do not develop spontaneous autoimmunity, Ikaros-deficient Treg are unable to control conventional T cell-mediated immune pathology in response to TCR and inflammatory stimuli in models of IBD and organ transplantation. These studies establish Ikaros as a core factor required in Treg for tolerance and the control of inflammatory immune responses.

## eLife assessment

This comprehensive study provides **valuable** information on the cooperation of Ikaros with Foxp3 to establish and regulate a major portion of the epigenome and transcriptome of T-regulatory cells. While the data are **compelling**, the evidence that these features are solely intrinsic, independent of the micro-environment, could be strengthened.

## Introduction

The zinc finger DNA binding protein Ikaros is expressed in hematopoietic precursors, where it regulates genes involved in antigen receptor recombination, hemoglobin synthesis, and genome stability by recruiting chromatin remodeling complexes (*Georgopoulos, 2002*). Germline deletion of *Ikzf1* in mice results in arrested erythroid and lymphoid development, leading to immunodeficiency and immature B and T cell leukemia. In conventional CD4+ and CD8+ T cells, Ikaros functions as a transcriptional repressor of inflammatory cytokine genes (*Thomas et al., 2007*; *Bandyopadhyay et al., 2012*; *Thomas et al., 2010*; *O'Brien et al., 2014*). Conventional CD4+ T cells with loss of Ikaros function are unable to differentiate into peripherally-induced regulatory T cells (iTreg) (*Shin et al., 2014*), and are resistant to suppression by thymic regulatory T cells (*Shin et al., 2014*). However, the role of Ikaros in thymic Treg development and function has not been addressed.

The Ikaros family members Helios and Eos have each been deleted or knocked down in human or mouse regulatory T cells (*Getnet et al., 2010*; *Pan et al., 2009*; *Gokhale et al., 2019*). Loss of Eos function in Treg is associated with increased expression of inflammatory cytokines like IL-2 and IFNg, and an inability to control pathogenic conventional T helper cell responses in an IBD model (*Pan et al., 2009*; *Gokhale et al., 2019*), although a separate study found that Eos-deficient Treg had normal function (*Rieder et al., 2015*). Helios also contributes to the control of Treg activation and cytokine production (*Sebastian et al., 2016*), but this may be secondary to its role in promoting stable expression of the *Foxp3* gene (*Getnet et al., 2010*; *Sebastian et al., 2016*; *Kim et al., 2015*).

In this study, we conduct a genome-scale multi-omic analysis of open chromatin, active histone marks, Ikaros occupancy, Foxp3 occupancy, and gene expression in wild-type and *Ikzf1*-deficient regulatory T cells. We find that Ikaros plays a crucial role in establishing the normal landscape of enhancer activity, Foxp3 binding, and gene expression in Treg that cannot be filled by other Ikaros family members. Loss of Ikaros function in Treg results in uncontrolled T cell-dependent inflammatory responses in vivo.

## Results

To address the role of Ikaros in the Treg lineage, we crossed mice with a floxed *Ikzf1* allele (*Schwickert et al., 2014*) with mice carrying a Foxp3-YFP-Cre reporter/driver. This strain generates an *Ikzf1*-null allele and the Cre neither generates a Foxp3 fusion protein nor affects Foxp3 function (*Rubtsov et al., 2008*). Male *Ikzf1*-fl-Foxp3-YFP-Cre mice (B6 background) do not develop overt autoimmune pathology under specific pathogen-free housing conditions over an 8-week timeframe, and basic aspects of T cell and Treg thymic development are unaltered (*Figure 1—figure supplement 1a*). CD4+CD25+Foxp3+peripheral Treg in these mice exhibit a nearly complete loss of Ikaros protein (*Figure 1a and b*) and full DNA demethylation at the *Foxp3* CNS2-TSDR (*Figure 1—figure supplement 1b*), indicating that they are of thymic origin (*Floess et al., 2007*). The expression of Ikaros in conventional CD4+ T cells from *Ikzf1*-fl-Foxp3-YFP-Cre mice is indistinguishable from that of control Foxp3-YFP-Cre mice (*Figure 1b*). *Ikzf1*-fl-Foxp3-YFP-Cre mice exhibit a statistically significant increase in total and effector Treg pools, with a concomitant decrease in the naive Treg pool in the spleen and lymph nodes (*Figure 1—figure supplement 1c*). *Ikzf1*-deficient Treg express normal levels of Foxp3, GITR, and PD1 (*Figure 1c and d*), and higher levels of the high-affinity IL-2 receptor CD25 and the costimulatory receptor ICOS (*Figure 1c*). Ikaros-deficient Treg maintained Eos, Aiolos, and Helios protein expression, exhibiting a mild increase in the expression of Eos and Aiolos particularly in the lymph nodes (*Figure 1d*).

### Ikaros contributes significantly to the Treg gene expression program

Ikaros is a transcription factor, so to assess how loss of Ikaros function impacts the Treg gene expression program, we compared the transcriptomes of wild-type and Ikaros-deficient Treg isolated directly ex vivo from 6 to 8 week old aged-matched mice (n=3). A total of 661 genes were differentially expressed in *Ikzf1* cko Treg, 149 of which greater than twofold (FDR <0.05, *Supplementary file 1*). Some of these genes were downregulated compared to wild-type Treg (*Figure 1e*, clusters 1, 4, and 5), but 80% were upregulated (*Figure 1e*, clusters 7 and 8), indicating that Ikaros functions primarily as a transcriptional repressor in Treg. Loss of Ikaros results in up-regulation of at least 12 factors that negatively regulate Treg function, e.g., multiple genes involved in Wnt signaling (*Wisp1*, *Ctnnd1*, *Ctnna1*), *Ox40*, *Tlr2*, *Lag3*, *Tnf*, and *Ifng* (*Figure 2a* and *Supplementary file 2*). Increased *Ifng* expression correlated with decreased DNA methylation at the *Ifng* locus compared to wild-type Treg (*Figure 1—figure supplement 1d*). *Ikzf1* cko Treg also exhibited down-regulation of at least 10 factors that are required for full Treg function, e.g., the activin receptor *Acvr1b*, *Nr4a1/Nur77*, *Tet1*, and perforin (*Figure 2b* and *Supplementary file 2*). *Bcl6*, which is required for follicular Treg function (*Nurieva et al., 2009*), is also down-regulated in Ikaros-deficient Treg. However, *Ikzf1* cko Treg up-regulated at least 24 factors known to promote Treg function (*Supplementary file 2*), suggesting that the loss-of-function program may be counteracted by a gain-of-function program in the absence of inflammation.

To simulate antigen encounters by Treg during an immune response, we stimulated wild-type and Ikaros-deficient Treg through the TCR and CD28 in vitro. Previous studies established a role for Ikaros

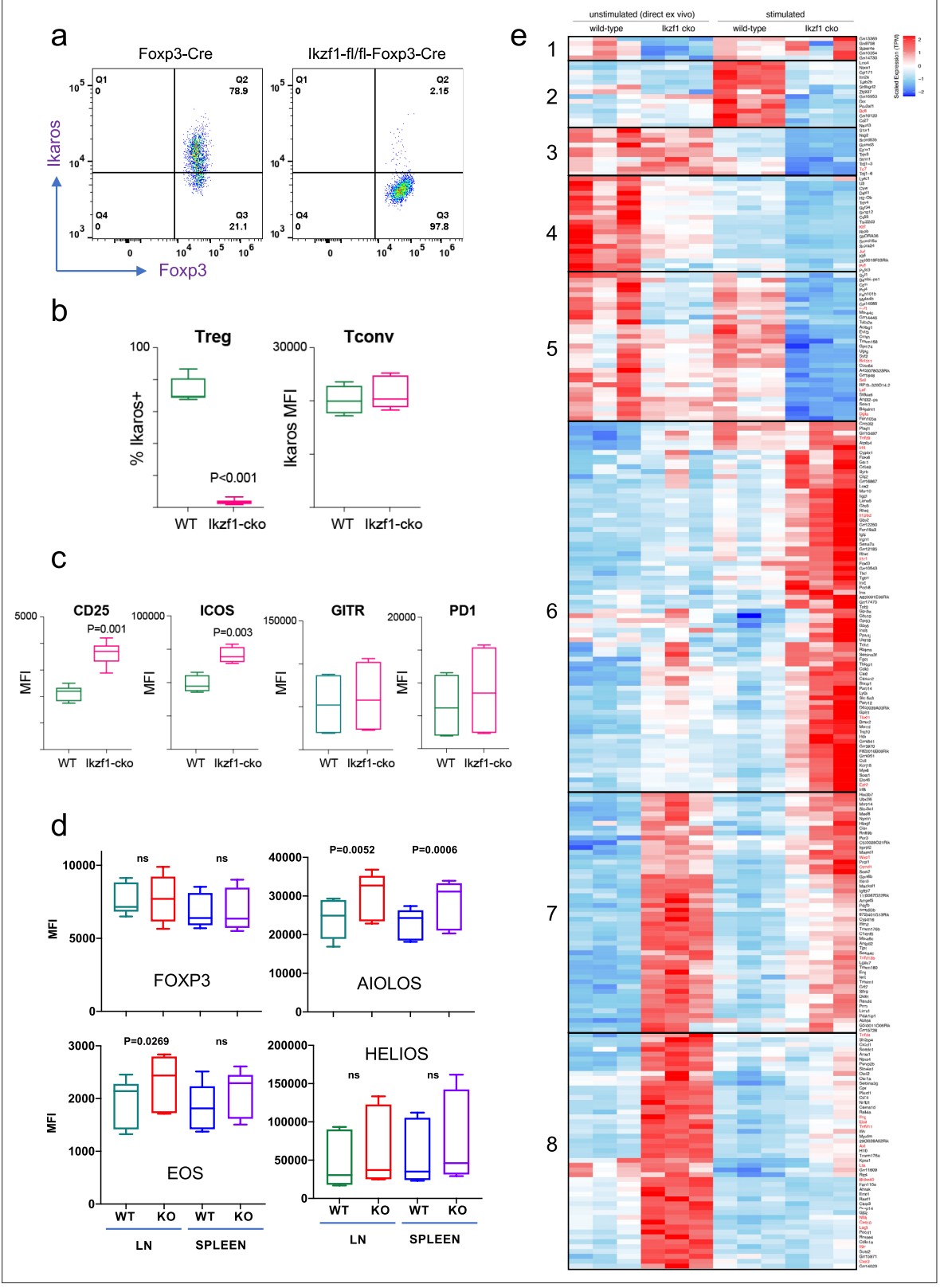

**Figure 1.** Impact of loss of Ikaros function on peripheral regulatory T cells (Treg) phenotype. Example histograms (**a**) and quantified expression (**b**) of Ikaros protein by peripheral Treg and Tconv from Foxp3-YFP-Cre (green) and *Ikzf1*-fl-Foxp3-YFP-Cre (red) mice (n=6 mice per group). (**c**) Expression of CD25, ICOS, GITR, and PD1 by wild-type (WT) (green) and *Ikzf1*-cko (red) Treg (n=6 mice per group). (**d**) Flow cytometric measurement of Foxp3, Aiolos, Eos, and Helios protein expression in WT and *Ikzf1*-cko Treg (n=6 mice per group). (**e**) Transcriptomic analysis of WT vs. *Ikzf1*-cko Treg gene expression.

*Figure 1 continued on next page*

Figure 1 continued

Top differentially expressed genes (FDR <0.05) organized into eight clusters in ex vivo or in vitro stimulated WT and *Ikzf1*-cko Treg. The heatmap represents scaled transcripts per million (tpm, n=3 replicates per group).

The online version of this article includes the following figure supplement(s) for figure 1:

**Figure supplement 1.** Immunophenotyping and DNA methylation analyses.

in restricting IL-2 and IFNg production by conventional CD4 + and CD8+ T cells (*Thomas et al., 2007*; *Bandyopadhyay et al., 2012*; *Thomas et al., 2010*; *O'Brien et al., 2014*). We find that Ikaros plays a similar role in Treg, as unlike wild-type Treg, *Ikzf1* cko Treg are capable of secreting IL-2 and IFNg protein upon stimulation (*Figure 3a*). Treg from mice expressing a dominant-negative form of Ikaros likewise ectopically express IL-2, IFNg, and TNFa at the protein level upon stimulation (*Figure 3b*). Consistent with their increased expression of IL-2 and IL-2R, Ikaros-deficient Treg exhibit enhanced IL-2-induced STAT5 phosphorylation (*Figure 3c*) and increased proliferative capacity (*Figure 3d*) compared to wild-type Treg.

At genome scale, stimulation through the TCR and CD28 led to differential expression of 895 genes (FDR <0.05, *Supplementary file 1*), 533 down-regulated (*Figure 1e*, clusters 2, 3, and 5), and 362 up-regulated in *Ikzf1* cko Treg compared to wild-type Treg (*Figure 1e*, clusters 6, 7, and 8). Consistent with its role in antigen receptor selection during lymphocyte development (*Georgopoulos et al., 1994*), gene ontology analysis (*Supplementary file 3*) of genes down-regulated in Ikaros-deficient Treg shows enrichment for immune cell development and TCR/VDJ recombination and diversification (*Figure 3e*). Genes upregulated in Ikaros-deficient Treg are enriched for networks involved in interferon and cytokine production and responses (*Figure 3e*). The set of upregulated genes includes at least 20 factors known to promote Treg function like *Foxp3*, *Il2ra*, *Icos*, *Ezh2*, and *Gpr83* (*Supplementary file 2*). However, loss of Ikaros results in up-regulation of at least 11 factors that negatively regulate Treg function like *Irf1*, *Il12rb2* (IL-12 receptor), *Il3*, and several genes in the Notch and Wnt pathways like *Notch2*, *Maml3*, *Rbpj*, *Wisp*, and *Ctnnd1* (*Figure 2c*), and down-regulation of at least 14 factors that are required for full Treg function like *Tcf7*, *Satb1*, *Foxp1*, *Id3*, *Smad3*, *Il27ra*, *Tlr7*, and follicular Treg (Tfreg) genes like *Bcl6*, *Cxcr5*, and *S1pr1* (*Figure 2d*).

The Wnt pathway is normally repressed in Treg, and ectopic Wnt signaling in Treg has been associated with ectopic IFNg production and reduced suppressive function (*Sumida et al., 2018*). We show that the Wnt signaling factor B-catenin is elevated in *Ikzf1* cko Treg (*Figure 4a*), and that ectopic IFNg production by Ikaros-deficient Treg is Wnt-dependent, while IFNg production by conventional CD4+ T cells does not depend on Wnt (*Figure 4b and c*). Together, these transcriptomic analyses indicate that Ikaros normally regulates a large proportion of the core Treg program (*Shevyrev and Tereshchenko, 2019*; *Figure 3f*, genes in red), and is required to restrain Wnt, Notch, and inflammatory cytokine gene expression in the Treg lineage. The impact of the loss of Ikaros function on the Treg transcriptome likely stem from both direct, cell-intrinsic effects and from indirect, cell-extrinsic effects.

## Ikaros is required for establishing the Treg open chromatin and enhancer landscape

To gain mechanistic insight into how Ikaros regulates the Treg gene expression program, we measured Ikaros binding, Foxp3 binding, chromatin accessibility, and H3K27ac enhancer marks in wild-type and Ikaros-deficient Treg using ATAC-seq and ChIP-seq. Loss of Ikaros function induced remodeling of 1431 genomic regions (n=3, FDR <0.05, *Figure 5—figure supplements 1 and 2* and *Supplementary file 4*), one-third of which (513) exhibiting reduced accessibility and two-thirds (918) of which become more accessible (*Figure 5a*). Regions with reduced accessibility in Ikaros-deficient Treg were enriched for nearby genes involved in leukocyte development and differentiation, while regions showing increased accessibility were enriched for genes involved in cytokine signaling and response to interferon-gamma (*Figure 5—figure supplement 1d*). At genome scale, increased accessibility at genomic elements after deletion of Ikaros correlated significantly with increased expression of nearby genes, while decreased accessibility correlated significantly with reduced gene expression (*Figure 5—figure supplement 1e*).

To explore how Ikaros regulates the Treg enhancer landscape, we measured histone acetylation at nucleosomes flanking open chromatin regions in wild-type and *Ikzf1* cko Treg (n=3, *Figure 5—figure*

**a** Negative regulators of Treg function UP in *ex vivo* Ikzf1 cko

| | |
|---|---|
| *Ctnna1* | WNT pathway opposes Treg function |
| *Tnfsf9/41BB* | opposes Treg suppression |
| *Bhlhe40* | promotes GMCSF expression |
| *Ifng* | subverts Treg stability & fxn |
| *Tnfsf4/Ox40L* | opposes Treg suppression |
| *Tnfrsf4/Ox40* | opposes Treg suppression |
| *Lag3* | limits Treg proliferation & function, inhibits Eos expr |
| *TNF* | opposes Treg suppression |
| *Klrg1* | marks term. diff. Treg with reduced prolif. & function |
| *Tlr2* | can push Treg to Th17 with reduced suppression |
| *Wisp1* | WNT, opposes Treg function |
| *Ctnnd1* | WNT, opposes Treg function |
| *Nfil3* | represses Foxp3 expr & Treg fxn |

**b** Positive regulators of Treg function DOWN in *ex vivo* Ikzf1 cko

| | |
|---|---|
| Tsc22d3/GILZ | cooperates with TGFB for induced Treg |
| Prf1/Perforin | cooperates with GZM in Treg suppression |
| Nr4a1/Nur77 | controls some Treg signature genes |
| Znrf3 | negative regulator of WNT |
| Vipr1 | vasoactive peptides induce Foxp3 |
| Gapdh | negative regulator of Ifng translation |
| Pdk1 | required for Treg maintenance |
| Fntb | promotes Treg maintenance |
| Acvr1b | activin A receptor, cooperates with TGFB for Foxp3 |
| Tet1 | promotes Foxp3/Treg stability |

**c** Negative regulators of Treg function UP in stim. Ikzf1 cko

| | |
|---|---|
| IL3 | induces NFIL3, which represses Foxp3 expr & Treg fxn |
| Il12rb2 | IL-12 signaling converts Th1-Treg to Th1 |
| Ppm1j | marker of unstable Treg |
| Irf1 | represses Foxp3 expression |
| Notch2 | opposes Treg function |
| Maml3 | Notch, opposes Treg function |
| Socs3 | Socs3 overexpr reduces FoxP3, CTLA-4, prolif., suppr. |
| Wisp1 | WNT, opposes Treg function |
| Ctnnd1 | WNT, opposes Treg function |
| Nfil3 | represses Foxp3 expr & Treg fxn |
| Tnfsf9/41BB | opposes Treg suppression |
| Tnf | opposes Treg suppression |

**d** Positive regulators of Treg function DOWN in stim. Ikzf1 cko

| | |
|---|---|
| Bcl6 | required for Treg/Tfr control Th2/Tfh resp. |
| Tcf7/TCF1 | control Treg/Tfr development/survival |
| S1pr1 | required for Treg accumulation at tumor/tissue sites |
| Cd27 | promotes expansion/accumulation of Treg |
| Ccr7 | required for Treg to supress in secondary lymph tissue |
| Smad3 | promotes Foxp3 induction downstream of TGFB |
| Cd5 | promotes Foxp3 induction by autoreactive Tconv |
| Traf3ip3 | required for Treg metabolic fitness and function |
| Cd28 | required for Treg-intrinsic homeostasis and function |
| Igf2r | signaling enhances TGFB release by Treg |
| Tlr7 | Tlr7 ligands enhance Treg suppression in vitro |
| Satb1 | required for Treg function |
| Foxp1 | critical regulator of Foxp3 and Treg function |
| Cxcr5 | contributes to Tfr GC migration |
| Il27ra | required for Treg control of inflammation |
| Id3 | Foxp3 expression, Treg stability, Tfr are Id3 low |

**Figure 2.** Survey of relevant differentially expressed genes in wild-type (WT) vs.*Ikzf1*-cko regulatory T cells (Treg). (**a**) Known negative regulators of Treg function up-regulated in ex vivo Ikzf1-cko Treg, (**b**) positive regulators of Treg function down-regulated in ex vivo Ikzf1-cko Treg, (**c**) negative regulators of Treg function up-regulated in in vitro stimulated Ikzf1-cko Treg, (**d**) positive regulators of Treg function down-regulated in in vitro stimulated Ikzf1-cko

*Figure 2 continued on next page*

*Figure 2 continued*

Treg. See ***Supplementary file 1*** for all differential genes and ***Supplementary file 2*** for a larger list of functionally relevant differentially expressed genes.

*supplement 2a and b*). Out of approximately 21,000 H3K27ac peaks called in both cell populations, 40% were affected by the loss of Ikaros function (***Supplementary file 5***), with 323 regions showing a >two-fold reduction acetylation and 1206 regions exhibiting a >two-fold increase in H3K27ac (***Figure 5b***). Differential analysis also showed that Ikaros controls histone acetylation at ~25% of Treg open chromatin regions (OCR, 9937 out of ~40,000, FDR <0.05), with the vast majority (78%, 7788) showing increased acetylation upon loss of Ikaros function (***Supplementary file 5*** and ***Figure 5— figure supplement 2c***). At genome scale, enrichment of the H3K27ac mark correlates with regional accessibility (***Figure 5—figure supplement 2d***), and strength of the enhancer signature at a given element correlates with the level of nearby gene expression (***Figure 5—figure supplement 2d***). Dense collections of multiple enhancers that tend to drive expression of genes involved in cell identity are called 'super-enhancers.' We defined 1700 Treg super-enhancers based on H3K27ac density (***Supplementary file 6***), 20% of which (324) are regulated by Ikaros (***Figure 5—figure supplement 2e***).

Ikaros ChIP-seq analysis identified 7642 Ikaros binding sites in WT Tconv, 83% of which (6361) are located in open chromatin, and 7061 Ikaros binding sites in WT Treg, 76% of which (5477) are located in open chromatin (***Figure 5c*** and ***Supplementary file 7***). Of all accessible Ikaros binding sites detected, 39% (3567) were Tconv-specific, 31% (2794) were shared between Tconv and Treg, and 29% (2683) were only detected in Treg (***Figure 5d***). The Ikaros ChIP-seq signal is enriched at accessible Ikaros motifs in the Treg and Tconv genome (***Figure 5e***), and the set of genes with accessible Ikaros binding motifs showed increased expression in Ikaros-deficient compared to wild-type Treg (***Figure 5f***). Motif analysis (***Supplementary file 8***) at regions that exhibit increased accessibility in *Ikzf1* cko Treg shows enrichment of the Ikaros GGGAA core binding sequence that is shared with immune trans-activators like NFkB, NFAT, Notch, and Stat1/4, and these regions are also enriched for AP-1 (Fos/Jun) and T-bet (Tbx21) motifs (***Figure 5g*** inset). This suggests that Ikaros can directly repress inflammatory gene expression in Treg by competing with NFkB, NFAT, Notch, and Stats for binding to enhancers and recruiting epigenetic factors that silence these elements (***Molnár and Georgopoulos, 1994***; ***Trinh et al., 2001***; ***Kleinmann et al., 2008***; ***Katerndahl et al., 2017***; ***Heizmann et al., 2020***). Loci under direct repressive control of Ikaros in Treg include *Bcl6*, *Notch2*, *Irf4*, and *Ifng* (***Figure 5h*** and ***Figure 5—figure supplement 3a–c***). However, the majority of genomic regions exhibiting increased accessibility in *Ikzf1* cko Treg are not bound by Ikaros in wild-type cells (873 of 918, ***Figure 5—figure supplement 3d***), suggesting that indirect gene regulation due to observed alterations in the expression of other transcription factors is another mechanism by which Ikaros establishes the Treg gene expression program.

## Foxp3 cooperates with Ikaros for DNA binding across the Treg genome

Regions that exhibit reduced accessibility in *Ikzf1* cko compared to wild-type Treg are enriched for the binding motif for Ikaros and the motif for TCF1 (***Figure 5g***), a factor that cooperates with Foxp3 to enforce Treg function (***Xing et al., 2019***) and is down-regulated in *Ikzf1* cko Treg (***Figure 3f***). These regions are likewise enriched for the GTAAACA Foxp3/forkhead motif (***Figure 5i*** inset), suggesting that Foxp3 may cooperate with Ikaros at these sites. To test this, we compared Foxp3 genome occupancy in wild-type vs. *Ikzf1* cko Treg by ChIP-seq (***Figure 6a***, ***Figure 6—figure supplement 1a–c***). A total of 4423 Foxp3 binding sites were detected in the open chromatin landscape of wild-type Treg (***Supplementary file 9***), and this ChIP-seq signal was enriched at accessible Foxp3 motifs. Consistent with the motif analyses (***Figure 5h***), we find a remarkable 74% of all Foxp3 binding sites are co-bound by Ikaros (3255 of 4423 sites, ***Figure 6b***). Loss of Ikaros in *Ikzf1* cko Treg results in reduced Foxp3 binding affinity at 70% of Ikaros-Foxp3 co-bound sites (2254 of 3256, ***Figure 6b*** inset), and reduced Foxp3 binding at 80% of all sites strongly bound by Foxp3 in wild-type Treg (3543 of 4422, ***Figure 6a and c***). As a result, the set of all direct target genes with accessible Foxp3 binding motifs showed significantly increased expression in Ikaros-deficient compared to wild-type Treg (***Figure 6d***).

Consistent with these observations, we find that Ikaros and Foxp3 exist in a complex in the nuclei of cells (***Figure 6e***), confirming prior proteomic data suggesting a physical interaction between Foxp3

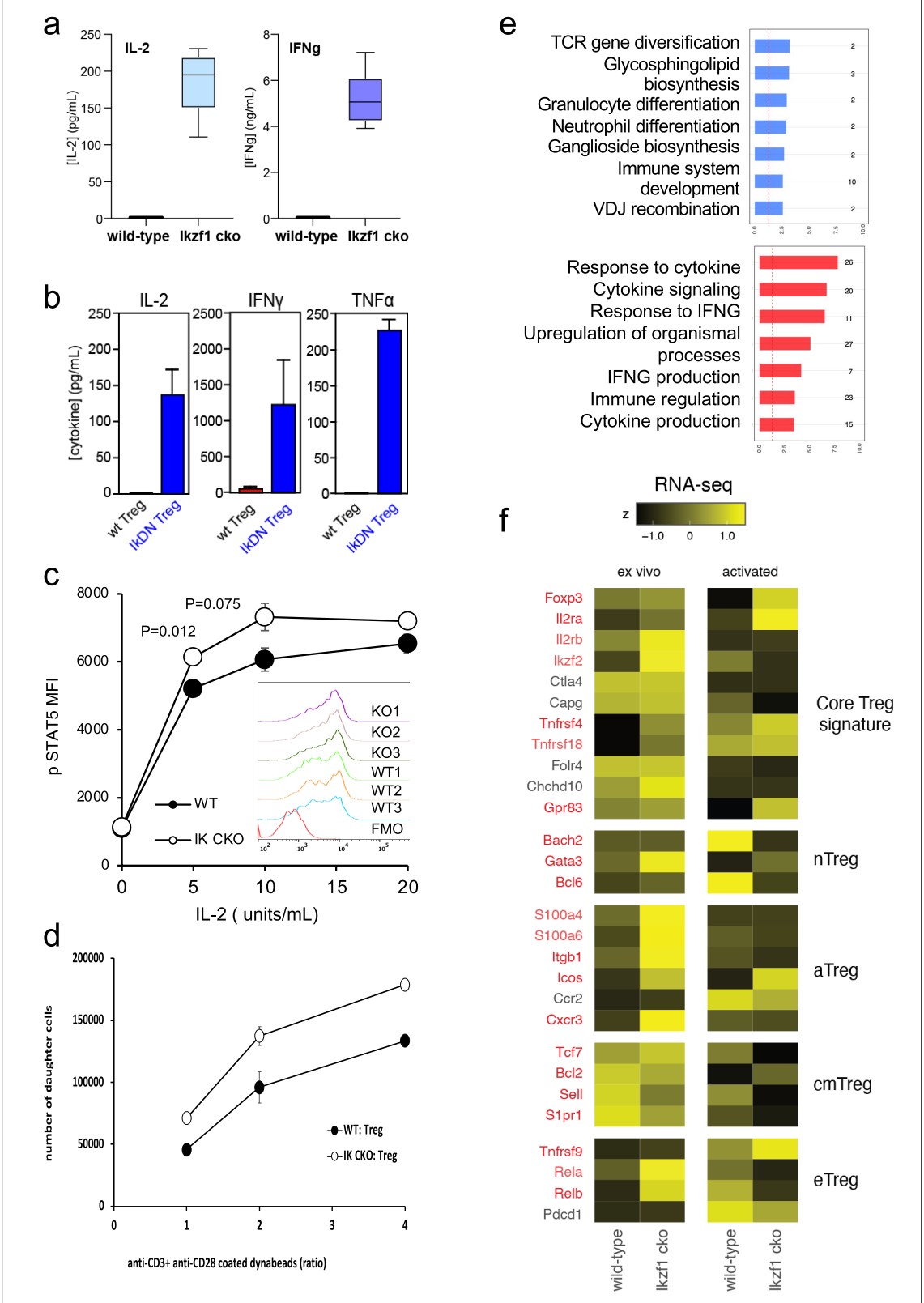

**Figure 3.** In vitro analysis of wild-type (WT) vs. *Ikzf1*-cko regulatory T cells (Treg). (**a**) Secretion of IL-2 and IFNg protein by WT and *Ikzf1*-cko Treg measured by ELISA (n=3). (**b**) IL-2, IFNg, and TNFa production by Treg from WT mice vs. mice with a dominant-negative form of Ikaros (IkDN) measured by ELISA (n=3). (**c**) IL-2-induced phosphorylation of STAT5 in WT and *Ikzf1*-cko Treg measured by flow cytometry in vitro. Mean fluorescence intensity (MFI) and individual histograms (inset, n=3) are depicted. (**d**) Activation-induced proliferation of WT (closed) and *Ikzf1*-cko (open) Treg measured by

*Figure 3 continued on next page*

*Figure 3 continued*

dye dilution (n=3). (**e**) Gene ontology analysis of genes down-regulated (top panel) and up-regulated (bottom panel) in *Ikzf1*-cko compared to WT Treg. The x-axis is fold enrichment and numbers to the right are unique genes in each pathway. (**f**) Differential expression of core Treg, nTreg, aTreg, cTreg, and eTreg genes (*Shevyrev and Tereshchenko, 2019*) in ex vivo (left panel) and in vitro stimulated (right panel) WT and *Ikzf1*-cko Treg. The heatmap represents z-score and genes significantly differentially expressed are shown in red.

and Ikaros in transfected cells (*Rudra et al., 2012*). At the Foxp3 and Ikaros co-bound target genes *Il2* and *Ifng*, retroviral expression of Foxp3 in CD4 + T cells results in direct promoter occupancy and silencing of both genes (*Figure 6f*). Co-expression of dominant-negative Ikaros (IkDN or Ik7) abrogates the binding of Foxp3 to the *Il2* promoter and inhibits the repressive activity of Foxp3 (*Figure 6f*). At genome scale, a significant number of genes in addition to *Il2* are concordantly regulated by both dominant-negative Ikaros and *Ikzf1* gene deletion (*Figure 6g*). In addition to the promoter, we also observe Ikaros-Foxp3 co-binding at a defined distal enhancer of *Il2* located 83 kb upstream of the promoter (*Mehra and Wells, 2015*) in wild-type Treg (*Figure 6h*). Loss of Ikaros function in *Ikzf1* cko Treg results in loss of Foxp3 binding at this enhancer, which is accompanied by increased

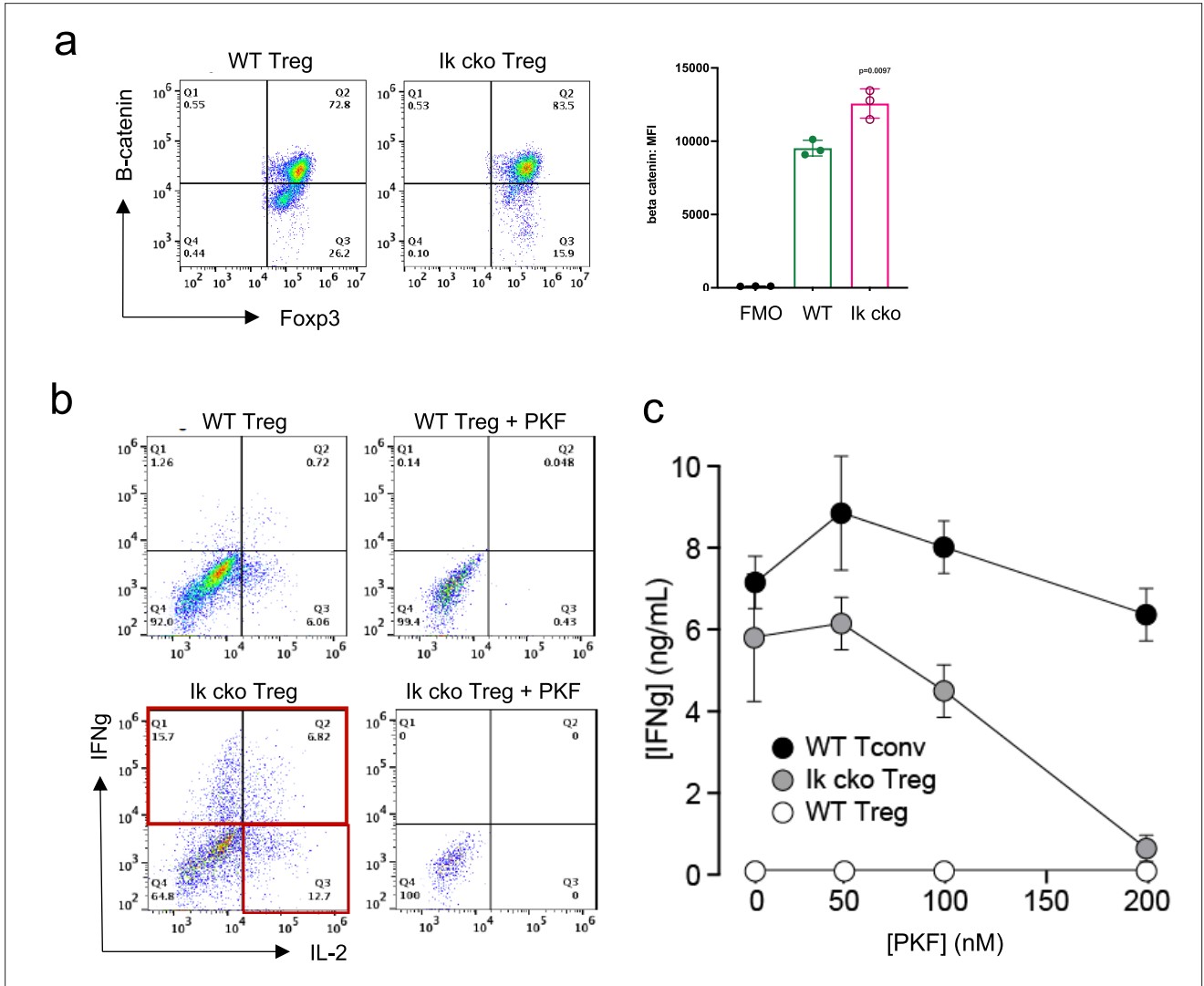

**Figure 4.** Ectopic activation of the Wnt-catenin pathway in *Ikzf1*-cko regulatory T cells (Treg). (**a**) Expression of B-catenin by wild-type (WT) (left histogram) vs. *Ikzf1*-cko (right histogram) Treg measured by flow cytometry (plot depicts B-catenin mean fluorescence intensity (MFI) from n=3 experiments). (**b**) IFNg secretion by WT (top panels) vs. *Ikzf1*-cko (bottom panels) Treg activated with (right panels) or without (left panels) the Wnt pathway inhibitor PKF. (**c**) IFNg secretion (measured by ELISA) by Ikzf1-cko Treg, but not by conventional T cells, is inhibited in a dose-dependent manner by PKF (n=3).

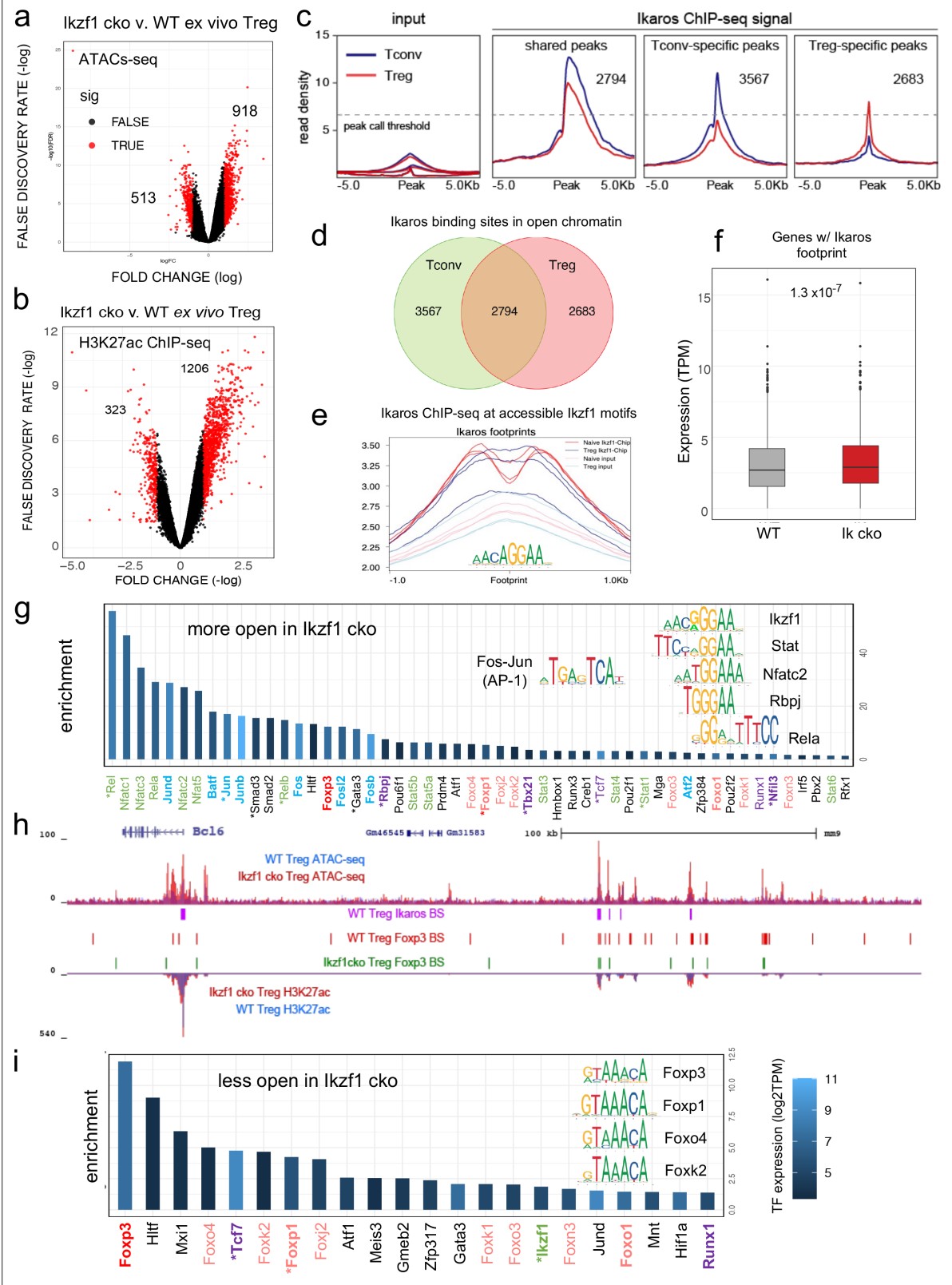

**Figure 5.** Genome-wide assessment of open chromatin, histone acetylation, and Ikaros occupancy in wild-type (WT) and *Ikzf1*-cko regulatory T cells (Treg). Differential analysis of open chromatin (**a**) and H3K27ac (**b**) in WT vs. *Ikzf1*-cko Treg (FDR <0.05, n=3). Peaks with FC >2 are depicted in red. (**c**) Ikaros ChIP-seq signal (read density) at genomic regions shared (panel 2) in Tconv (blue) vs. Treg (red), unique to Tconv (panel 3), or unique to Treg (panel 4). Panel 1 depicts read densities at the same regions in control input libraries. (**d**) Unique vs. shared Ikaros binding sites in Treg (red) vs. Tconv

*Figure 5 continued on next page*

*Figure 5 continued*

(green) open chromatin. (**e**) Enrichment of Ikaros ChIP-seq signal (footprint, solid lines) or input background (transparent lines) at accessible Ikaros motifs (AGGAA) in WT Treg (blue) and Tconv (red). (**f**) Expression (tpm) of genes with open chromatin enriched for the Ikaros consensus binding motif in WT vs. *Ikzf1*-cko Treg. (**g**) Enrichment of TF consensus binding motifs in genomic regions that are more accessible in *Ikzf1*-cko Treg. Inset depicts motifs for Ikaros/Ikzf1, Stat, Nfat, Rbpj, Rela, and AP-1. (**h**) Open chromatin (top tracks) and H3K27ac (bottom tracks) in WT (blue) and *Ikzf1*-cko (red) Treg, Ikaros binding sites (purple marks) and Foxp3 binding sites (red marks) in WT Treg, and Foxp3 binding sites in *Ikzf1*-cko Treg (green marks) at the *Bcl6* locus. (**i**) Enrichment of TF consensus binding motifs in genomic regions that are less accessible in *Ikzf1*-cko Treg. Inset depicts motifs for forkhead family members. In (**g**) and (**i**), factors with roles in Treg function are colored green and purple, forkhead family members are colored red, and factors differentially expressed in *Ikzf1*-cko Treg are indicated with an asterisk. ATAC-seq was performed on Treg purified directly ex vivo, while ChIP-seq analyses were performed on Treg expanded in vivo using IL-2/anti-IL-2 complexes.

The online version of this article includes the following figure supplement(s) for figure 5:

**Figure supplement 1.** ATAC-seq and H3K27ac ChIP-seq library metrics.

**Figure supplement 2.** ATAC-seq and H3K27ac ChIP-seq library metrics.

**Figure supplement 3.** Multi-omic integration of differential ATAC-seq, ChIP-seq and RNA-seq analyses.

histone acetylation, chromatin accessibility (*Figure 6h*), and *Il2* expression (*Figures 1 and 2*). Other examples of Foxp3-regulated genes that exhibit Ikaros-dependent Foxp3 binding, enhancer activity, and expression are *Tcf7* (*Figure 6i*), *Il2ra, Rbpj,* and *Maml3* (*Figure 6—figure supplement 1d and e*). Together, these results indicate that a large portion of the Treg epigenome and transcriptome, including two-thirds of the core Treg program and the majority of the Foxp3 gene regulatory program, is dependent on Ikaros.

## Ikaros is required for Treg control of conventional T cell differentiation

The large-scale dysregulation of gene expression in *Ikzf1*-cko Treg, especially upon stimulation, suggests that extrinsic control of inflammatory immune responses may be dysregulated in mice lacking Ikaros in the Treg lineage. We observed no clear signs of frank autoimmunity in aged (1-year-old) *Ikzf1*-fl-Foxp3-YFP-Cre mice (*Figure 7—figure supplement 1a*). The conventional T cell pool in 6–8 week-old (*Figure 7*) and 10-month-old (*Figure 7—figure supplement 1*) *Ikzf1*-fl-Foxp3-YFP-Cre mice showed a statistically significant accumulation of CD4+ T cells (*Figure 7a*, *Figure 7—figure supplement 1b*) with a memory phenotype (*Figure 7b and c*), and a concomitant reduction in naive phenotype CD4+ T cells (*Figure 7d*, *Figure 7—figure supplement 1b*). Ikaros-deficient Treg maintained comparable suppressive activity against wild-type Tconv in vitro (*Figure 7e*), however, Ikaros-sufficient, conventional CD4+ T cells from mice lacking Ikaros in the Treg lineage were resistant to suppression by both *Ikzf1*-cko and wild-type Treg (*Figure 7f and g*), likely due increased frequency of suppression-resistant memory cells (*Yang et al., 2007*; *Afzali, 2011*). At 6 months, *Ikzf1*-fl-Foxp3-YFP-Cre mice also exhibited increased frequencies of follicular helper CD4 + T cells in the lymph nodes and spleen (*Figure 7h*), which was associated with elevated levels of total IgM, IgG, and especially IgA in the serum (*Figure 7i*). The elevated IgA was accompanied by increased frequencies of IgA-positive B cells in the spleen and mesenteric lymph nodes (*Figure 7j and k*). These results suggest perturbed immune homeostasis in mice that lack Ikaros function in Treg owing to a defect in the control of conventional CD4 + T cell differentiation.

## Ikaros is required for Treg control of pathogenic T cell-mediated mucosal inflammation and acquired immune tolerance

To address this, we tested the ability of Ikaros-deficient Treg to control inflammatory colitis in an in vivo adoptive transfer model of IBD. Rag-deficient mice that received conventional CD4+CD25-negative T cells alone (n=5) developed severe disease as evidenced by progressive weight loss (*Figure 8a*), gross and histological intestinal pathology (*Figure 8b* and *Figure 8—figure supplement 1a*), extensive cellular infiltration, and tissue damage in the inner mucosal epithelial layer of the colon (*Figure 8c*), and high numbers of activated Tconv in the colon (*Figure 8c*), spleen and mesenteric lymph nodes of these animals (*Figure 8—figure supplement 1d–g*). Upon co-transfer into Rag-deficient mice (n=5), wild-type Treg accumulated in the mesenteric lymph nodes (*Figure 8—figure supplement 1i*) and colon (*Figure 8d*), were able to control the activation and expansion of conventional helper T cells in lymphoid tissues (*Figure 8—figure supplement 1e and g*), and limit CD4+ T cell infiltration into the

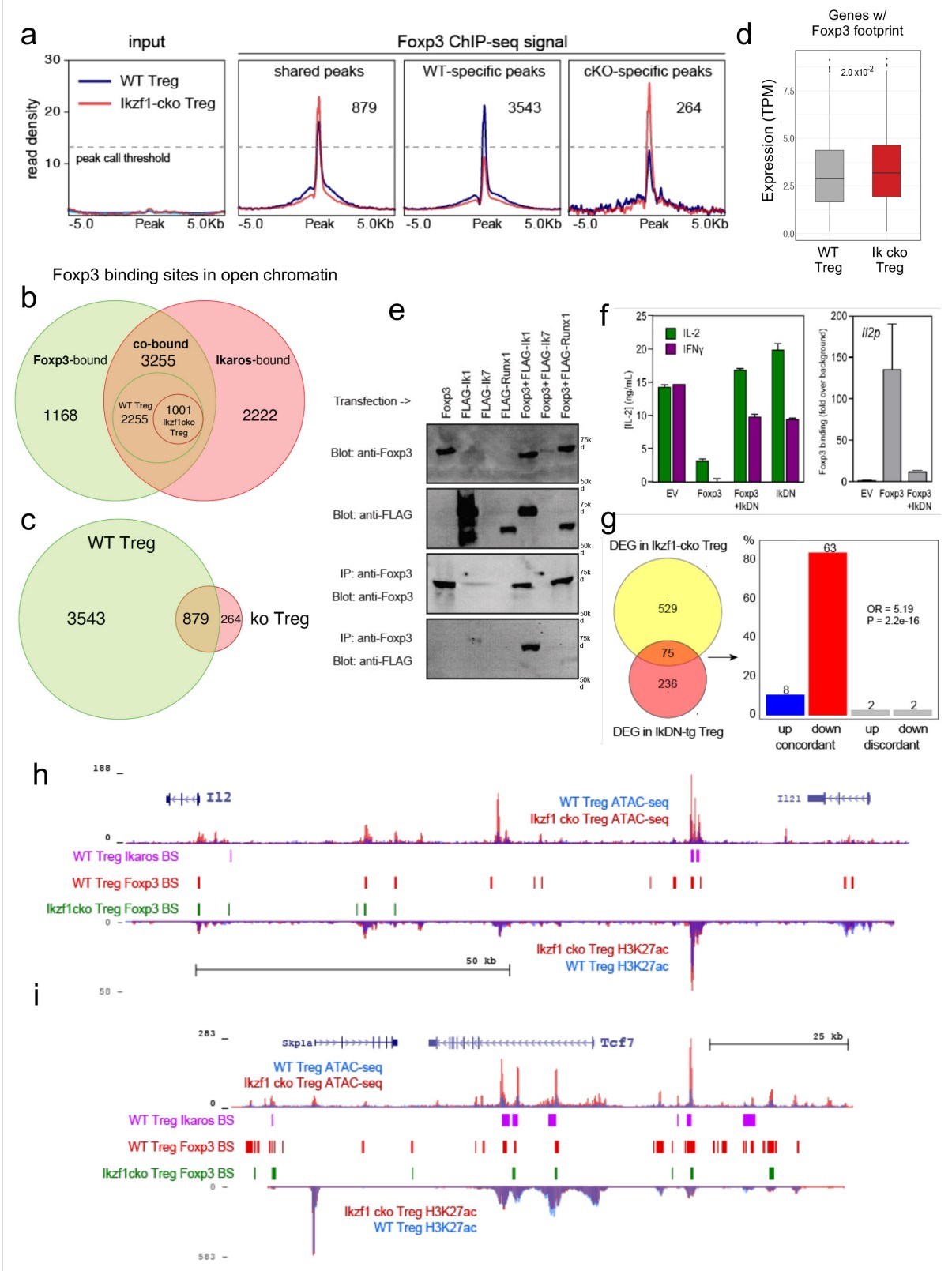

**Figure 6.** Ikaros-dependent Foxp3 function in regulatory T cells (Treg). (**a**) Input vs. Foxp3 ChIP-seq at genomic regions shared or unique in wild-type (WT) (blue) vs. *Ikzf1*-cko (red) Treg (n=3 per group). (**b**) Foxp3- (green), Ikaros- (red), and Foxp3-Ikaros co-bound (orange) open chromatin regions (OCR). Inset depicts Foxp3-Ikaros co-bound regions in WT (green) vs. *Ikzf1*-cko (red) Treg. (**c**) Foxp3 binding sites in WT (green) vs. *Ikzf1*-cko (red) Treg. (**d**) Expression (tpm) of genes enriched for accessible Foxp3 consensus motifs in WT vs. *Ikzf1*-cko Treg. (**e**) 293T cells transfected with FLAG-tagged

*Figure 6 continued on next page*

*Figure 6 continued*

full-length Ikaros (Ik1), DNA-binding mutant Ikaros (Ik7), or Runx1 alone (lanes 1–4) or in combination with untagged Foxp3 (lanes 5–7). Whole extracts (panels 1–2) or Foxp3-immunoprecipitated extracts (panels 3–4) probed for Foxp3 or FLAG. (**f**) IL-2 or IFNg production (left panel) and Foxp3 ChIP-qPCR at *Il2* promoter (right panel) in Tconv transduced with vector, Foxp3, Ik7/DN, or Foxp3 + Ik7/DN. (**g**) Concordant vs. discordant genes co-regulated in *Ikzf1*-cko vs. IkDN Treg (odds ratio = 5.19, *P*-value = $2.2 \times 10^{-16}$). (**h, i**) Open chromatin (top) and H3K27ac (bottom) in WT (blue) and *Ikzf1*-cko (red) Treg, Ikaros binding (purple marks), and Foxp3 binding (red marks) in WT Treg, and Foxp3 binding in *Ikzf1*-cko Treg (green marks) at *Il2* (**h**) and *Tcf7* (**i**).

The online version of this article includes the following source data and figure supplement(s) for figure 6:

**Source data 1.** Immunoblot analysis of Foxp3-Ikaros co-precipitation in transfected HEK293T cells.

**Figure supplement 1.** Foxp3 ChIP-seq library metrics.

intestinal epithelium (*Figure 8d*). Recipients of wild-type Treg exhibited only mild intestinal pathology at the gross and microscopic levels (*Figure 8b and d* and *Figure 8—figure supplement 1b*), and lost little weight over the course of the experiment (*Figure 8a*). Co-transferred Ikaros-deficient Treg accumulated in the spleen (*Figure 8—figure supplement 1h and i*) and, despite reduced expression of the alpha4-beta7 integrin (*Itga4, Itgb7*) and *Ccr9* genes involved in homing to the intestine, accumulated in the intestinal epithelium to numbers 5–10-fold higher than in the recipients of wild-type Treg (*Figure 8e*). Despite the presence of large numbers in the intestinal mucosa, Ikaros-deficient Treg were completely unable to protect RAG-deficient mice (n=5) from infiltration and colitis mediated by wild-type conventional T cells at the level of weight loss (*Figure 8a*) and intestinal pathology (*Figure 8b and e* and *Figure 8—figure supplement 1c*). Similar to mice that received no Treg, recipients of Ikaros-deficient Treg exhibited extensive inflammatory infiltrates in the colon, with thickening and detachment of the epithelial layer from underlying tissues (*Figure 8e*). Cell necrosis and mononuclear lymphocyte infiltration were also more pronounced in recipients of Ikaros-deficient Treg. These results indicate that Treg depend on Ikaros to control mucosal inflammation during a conventional T cell response.

Regulatory T cells are required for the induction of peripheral alloimmune tolerance. To determine whether Treg-intrinsic Ikaros function is required for acquired tolerance to organ transplants, we transplanted fully mismatched cardiac allografts into *Ikzf1*-fl-Foxp3-YFP-Cre or control Foxp3-YFP-Cre mice under combined blockade of the CD28 and CD40 costimulatory pathways (n=5 per group). While costimulatory blockade induced long-term allograft tolerance in wild-type recipients, this treatment failed to induce tolerance in mice lacking Ikaros in the Treg lineage (*Figure 9a*). Similar results were obtained when anti-CD40L plus donor-specific transfusion was used as a tolerizing regimen (*Figure 8—figure supplement 1j and k*). Intragraft analysis of gene expression showed elevated levels of multiple Th1-related transcripts in rejecting grafts from *Ikzf1*-Treg-cko recipients, despite elevation of Foxp3 (*Figure 9b*). Histopathological analysis of cardiac graft tissue from *Ikzf1*-Treg-cko recipients showed extensive myocardial necrosis (*Figure 9c*) associated with increased CD4 + T cell infiltration despite numbers of Foxp3+ Treg comparable to that in tolerant recipients (*Figure 9d*). These results reveal an important role for Ikaros in the ability of Treg to control inflammation and establish acquired immune tolerance.

## Discussion

The studies reported here establish a crucial role for Ikaros in regulatory T cells that cannot be replaced by the Ikaros family members Helios, Eos, or Aiolos. While Ikaros is required for induction of Foxp3 by TGF-B in conventional T cells (*Heller et al., 2014*), thymic-derived Ikaros-deficient Treg exhibited normal Foxp3 expression. Instead, loss of Ikaros activity results in significant dysregulation of the Treg gene expression program, including pro-inflammatory cytokine and chemokine genes normally not expressed by Treg (e.g. *Ifng, Tnf, Il3, Il12rb, Tlr2*), and genes required for normal Treg function (e.g. *Tcf7, Lef1, Satb1, Nr4a1*). Many of these dysregulated genes are Foxp3 targets, and we show that Foxp3 cooperates with Ikaros to occupy the majority of Foxp3 binding sites in Treg.

Mice with Treg lineage-specific loss of Ikaros occupancy showed an accumulation of activated regulatory and helper T cells in the secondary lymphoid tissues, but no evidence for increased T cell infiltration into organs or frank autoimmunity. This might be explained by the fact that Treg isolated directly ex vivo showed elevated expression of many genes that promote Treg function, potentially balancing the dysregulated program predicted to have deleterious effects on Treg homeostasis and function.

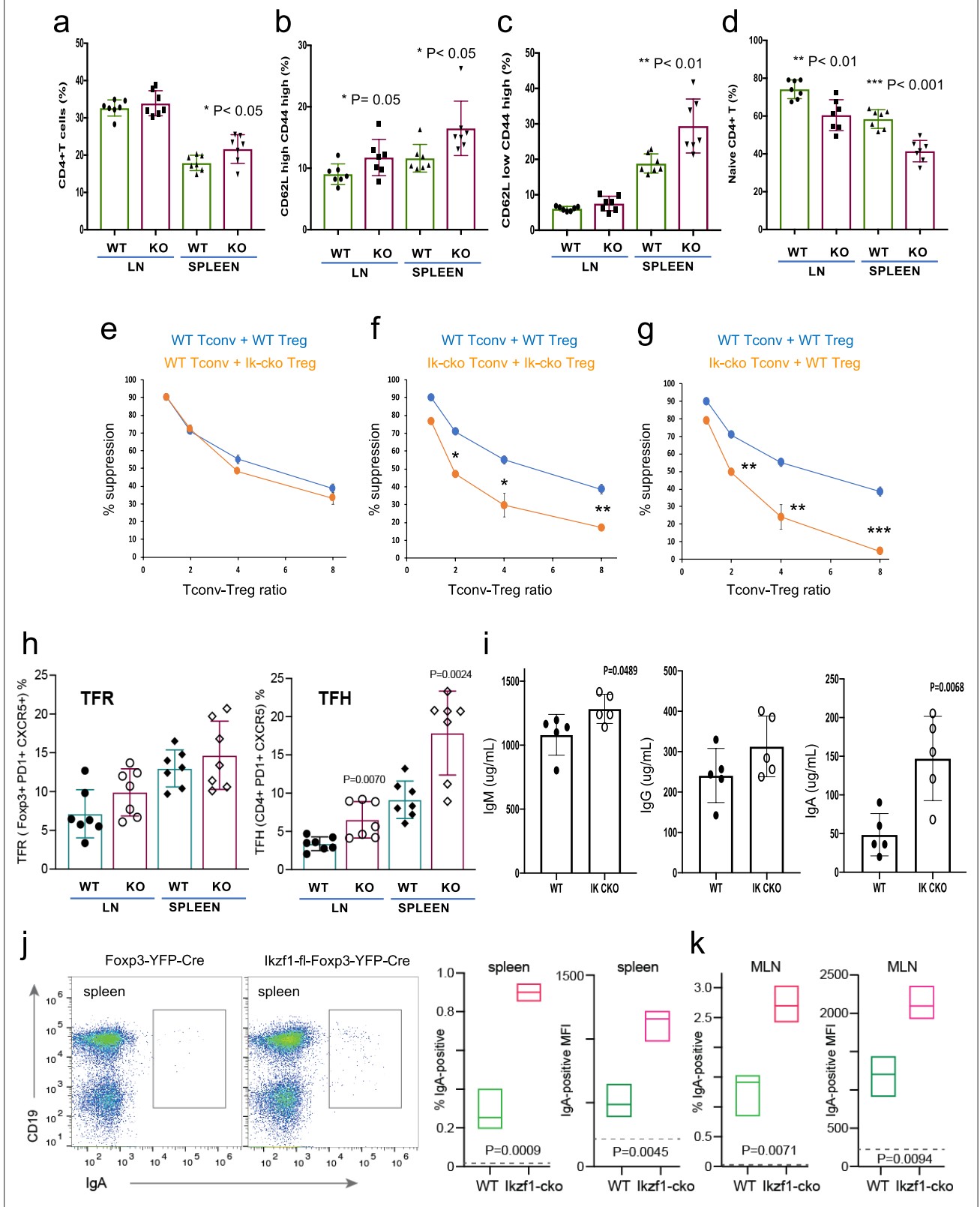

**Figure 7.** Immunophenotyping of Tconv from *Ikzf1*-fl-Foxp3-YFP-Cre and Foxp3-YFP-Cre mice. Frequencies of total (**a**), memory (**b, c**), and naïve (**d**) phenotype Tconv in secondary lymphoid tissues of 6–8 week-old *Ikzf1*-fl-Foxp3-YFP-Cre (purple) and Foxp3-YFP-Cre (green) mice (n=7). (**e**) In vitro suppressive activity of wild-type (WT) (blue) vs. *Ikzf1*-deficient (orange) Treg against Tconv from WT mice. (**f**) In vitro suppressive activity of WT Treg against Tconv from WT mice (blue) vs. *Ikzf1*-deficient Treg against Tconv from *Ikzf1*-cko mice (orange). (**g**) In vitro suppressive activity of WT Treg

*Figure 7 continued on next page*

*Figure 7 continued*

against Tconv from WT (blue) vs. *Ikzf1*-deficient mice (orange). Tconv proliferation was measured by dye dilution in all cultures (n=3). (**h**) Frequencies of Foxp3+PD1^hiCXCR5^hi follicular regulatory T cells (Tfr) and PD1^hiCXCR5^hi follicular helper T cells (Tfh) in secondary lymphoid tissues of 6-month-old *Ikzf1*-fl-Foxp3-YFP-Cre (purple) and Foxp3-YFP-Cre (green) mice (n=7). (**i**) Total serum levels of IgM, IgG, and IgA from *Ikzf1*-fl-Foxp3-YFP-Cre vs. Foxp3-YFP-Cre mice (n=4). Frequency of IgA-positive B cells and surface density (MFI) of IgA-positive B cells in spleen (**j**) and mesenteric lymph nodes (**k**) from 9-month-old WT (green) vs. *Ikzf1*-deficient (red) mice (N=3). p-values are indicated for significant differences.

The online version of this article includes the following figure supplement(s) for figure 7:

**Figure supplement 1.** Histology of 1-year-old *Ikzf1*-fl-Foxp3-YFP-Cre and Foxp3-YFP-Cre control mice.

However, upon TCR stimulation, *Ikzf1*-deficient Treg induce a set of inflammatory Th1, Notch, and Wnt pathway genes normally repressed in Treg, and fail to control in vivo cellular and humoral immune responses mediated by conventional T cells. Importantly, our results show that the loss of suppressive function in vivo is not due to failure of Treg to home to sites of tissue inflammation. The lack of spontaneous inflammation or autoimmunity in *Ikzf1*-fl-Foxp3-Cre mice is similar to mice with Treg-specific deletion of *Prdm1* (**Bankoti et al., 2017**; **Cretney et al., 2018**), *Icos* (**Guo et al., 2008**), *Il10* (**Rubtsov et al., 2008**) and *Mef2d* (**Di Giorgio et al., 2020**). Mice lacking *Il10* in Treg exhibit mild colitis, but no autoimmunity, while mice with deletion of *Prdm1* in Treg show signs of autoimmunity only in aged mice. Moreover, deletion of *Mef2d, Blimp1, Icos,* or *Il10* in Treg does not impact their suppressive activity in vitro, but impairs their function in vivo in the context of inflammation.

Although the literature is not in complete agreement, Eos and Helios are considered to be necessary for Treg function through their contribution to the Treg gene expression program. Eos, like Ikaros, is required to repress inflammatory gene expression by Treg (**Pan et al., 2009**; **Gokhale et al., 2019**), and Eos can cooperate with Foxp3 to strengthen a core Treg gene expression program when ectopically co-expressed in conventional T cells (**Fu et al., 2012**). Loss of Helios function in Treg has the primary effect of destabilizing *Foxp3* expression (**Getnet et al., 2010**; **Sebastian et al., 2016**; **Kim et al., 2015**), and also contributes to core Treg gene expression when ectopically co-expressed with Foxp3 (**Fu et al., 2012**). Despite these functions, neither Eos, Helios, nor Aiolos were able to compensate for the loss of Ikaros, despite the fact that *Ikzf1*-deficient Treg express all these proteins at comparable or higher levels as compared to wild-type Treg. The availability of Helios ChIP-seq (**Kim et al., 2015**) and Helios-dependent gene expression (**Yates et al., 2018**) data allowed us to compare Ikaros and Helios genome occupancy and gene regulatory programs (***Figure 10***). Of the 1838 Helios binding sites and 5477 Ikaros binding sites detected in Treg open chromatin, 64 are shared, representing only ~3.5% of Helios binding sites and ~1% of Ikaros binding sites. Similarly, we found a statistically significant overlap between the Ikaros-dependent set of 660 genes and the Helios-dependent set of 147 genes, but this consisted of only nine genes (***Figure 10***). For five of these genes, Ikaros and Helios have the same effect on expression, while Ikaros and Helios have the opposite effect on expression of the other four genes. This level of discordance at the level of both genome occupancy and gene regulation likely explains whey Helios, for example, cannot compensate for the loss of Ikaros in Treg.

Coding mutations in *IKZF1* in humans are a cause of common variable immune deficiency (CVID) and autoimmunity (**Boast et al., 2021**). To date, studies have focused on the impact of these mutations on T and B lymphocyte function, and our results here suggest Treg defects could also contribute to the immune dysregulation observed in these patients. In addition, common genetic polymorphism at the *IKZF1* locus have been associated with SLE susceptibility by GWAS (**Cunninghame Graham et al., 2011**), and one mechanism for this is an SLE-associated distal regulatory element required for normal expression of Ikaros in human T cells (**Su et al., 2020**). Given the important roles for Ikaros in conventional B and T cell function, and its role newly defined here in regulatory T cell function, Ikaros is a relevant target for novel therapies for autoimmunity, organ transplant rejection, and cancer.

## Methods
### Antibodies
Fluorochrome-conjugated anti-mouse monoclonal antibodies CD3-AF700 (cat # 100216), CD4-BV785 (cat # 100453), CD8-PB (cat # 100725), CD25-BV650 (cat # 102038), GITR-PECy7 (cat # 120222), ICOS-PECy5 (cat # 107708), IFNg-PeCy7 (cat # 505826), and IL-2-PB (cat # 503820) were purchased from Biolegend. CD44-Percp-cyanine5.5 cat# 45-0441-80, CD62L-APCeFL780 (cat # 47-0621-82),

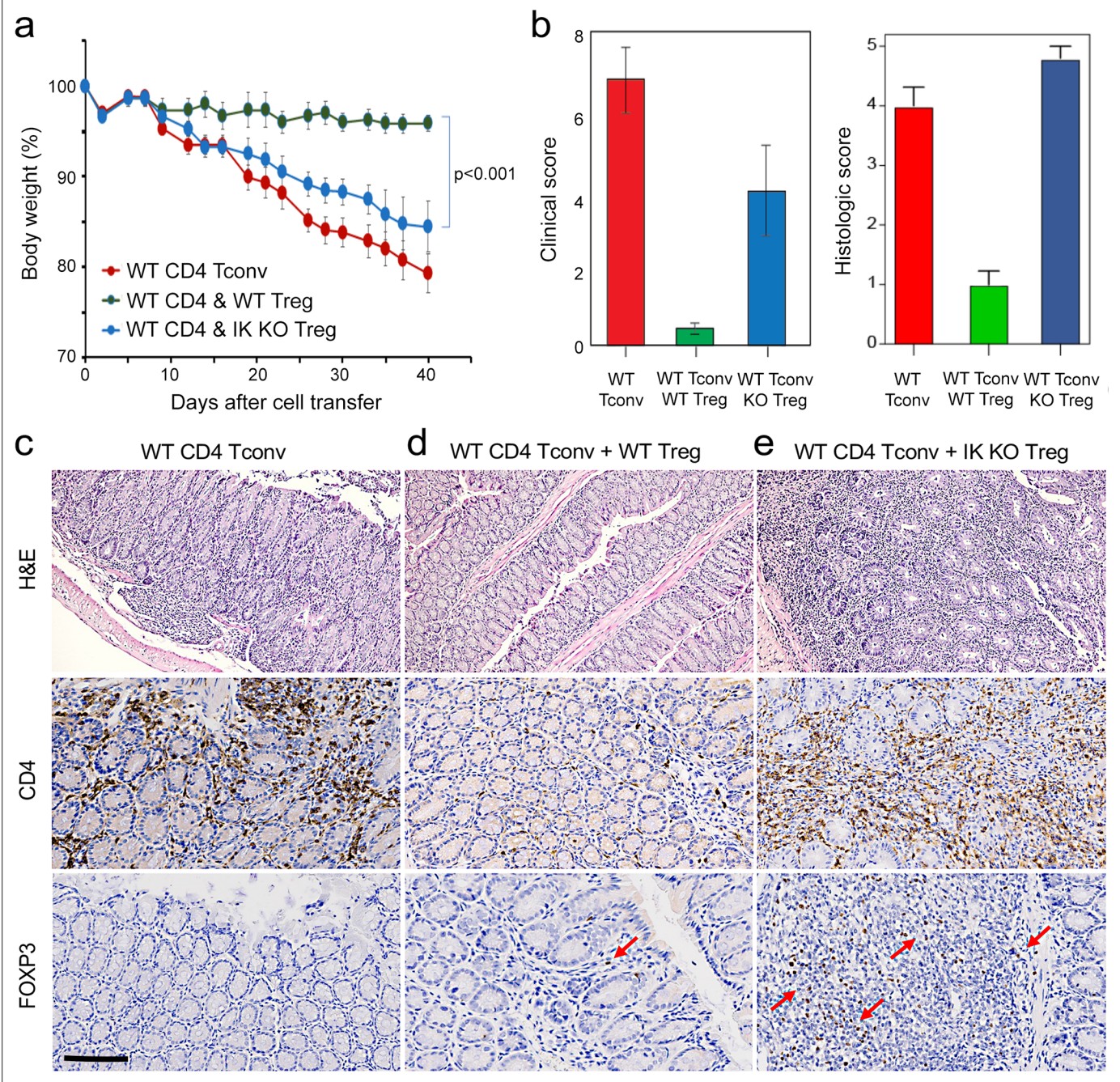

**Figure 8.** Role of Ikaros in regulatory T cells (Treg)-mediated control of inflammatory colitis. Wild-type (WT) CD4 +CD25- Tconv were transferred alone (red), or together with WT (green) or *Ikzf1*-cko (blue) CD4 +CD25+Treg into RAG1ko mice (n=5). Animal weight was monitored for 40 days (**a**), and intestines were scored for pathology at the gross and histologic levels (**b**). Example histopathology of colons from RAG1ko recipients of WT Tconv (**c**), WT Tconv+WT Treg (**d**), and WT Tconv+*Ikzf1* cko Treg (**e**). Hematoxylin and eosin (H&E) (top row), CD4 (middle row), and Foxp3 (bottom row) staining are shown at 200x. Scale bar = 100 μm. Mean Foxp3 + cells per 200 X field from n=3 animals is 3.4 in (**d**) and 22 in (**e**), *p*<0.05.

The online version of this article includes the following figure supplement(s) for figure 8:

**Figure supplement 1.** Role of Ikaros in regulatory T cells (Treg)-mediated protection from colitis.

Foxp3-APC (cat# 17-5773-82), Eos-eFL660 (cat # 50-5758-80), Helios-PeCy7 (cat # 25-9883-42), Aiolos-PE (cat # 12-5789-80),and bCatenin-eFL660 (cat # 50-2567-42) were purchased from Thermo Fisher Scientific. PD-1-PECy7 (cat# 25-9985-80), CXCR5-BV421 (cat # 562889), Bcl-6-PE (cat # 569522), and phospho-STAT5-PE (pY694) (cat # 612567) were procured from BD Biosciences.

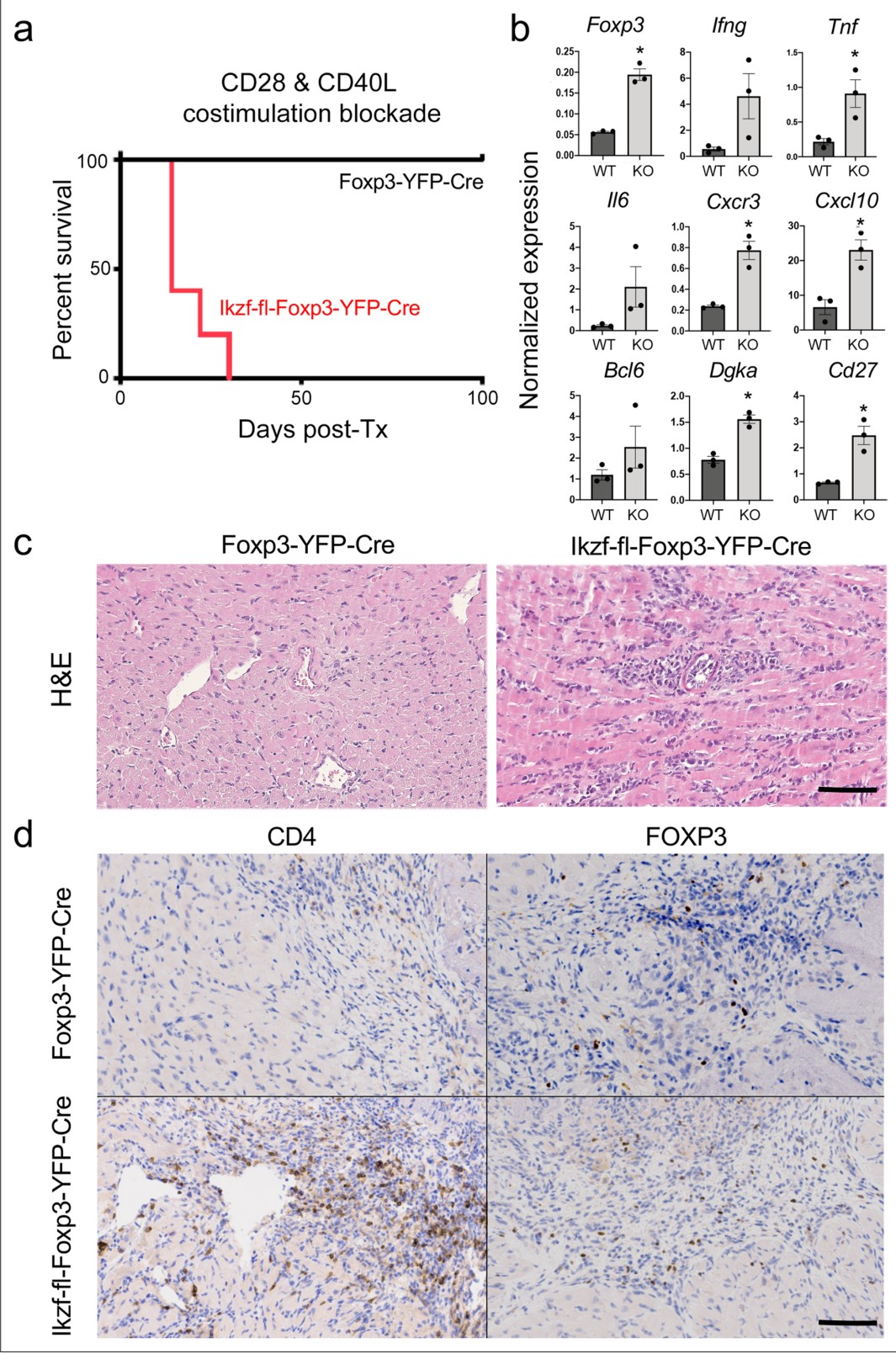

**Figure 9.** Role of Ikaros in regulatory T cells (Treg)-dependent acquired cardiac transplant tolerance. (**a**) B6 *Ikzf1*-fl-Foxp3-YFP-Cre (red) or Foxp3-YFP-Cre (black) mice (n=5) received BALB/c cardiac allografts under combined CD28 +CD40 costimulatory blockade and graft survival was monitored for 100 days. (**b**) Analysis of intra-graft transcript levels of the indicated genes from grafts harvested at day 19 post-transplant (*p<0.01). Histopathological

*Figure 9 continued on next page*

*Figure 9 continued*

analysis of cardiac grafts harvested at day 19 post-transplant from Foxp3-YFP-Cre and *Ikzf1*-fl-Foxp3-YFP-Cre recipients by hematoxylin and eosin (H&E) (**c**) and CD4 and Foxp3 (**d**) staining (scale = 200 x, scale bar = 100 μm).

## Mice

The conditional *Ikzf1*-fl/fl mouse was provided by Dr. Meinrad Busslinger (*Schwickert et al., 2014*). Mice were maintained at the Department of Veterinary Resource facility of CHOP. All animal experiments were performed according to protocols and guidelines approved by the CHOP animal care and use committee. To generate mice with conditional deletion of Ikaros in Tregs, homozygous *Ikzf1*-fl/fl mice were crossed with Foxp3-IRES-YFP-Cre mice (*Rubtsov et al., 2008*) purchased from JAX (Strain #016959). Specific deletion of *Ikzf1* in Treg was confirmed by flow staining.

## Immunophenotyping and ELISA

Thymus, spleen, and lymph nodes were collected from individual mice (5–7 weeks old) and single-cell suspension was prepared in 1X PBS. RBC lysed lymphocytes were stained for CD3, CD4, CD8, CD25, CD44, CD62L, GITR, PD-1, ICOS, Foxp3, and Ikaros. For intracellular transcription factor staining, cells were fixed with eBioscience Perm/fix buffer (Thermo Fisher Scientific). To determine Ikaros and Foxp3 expression, permeabilized cells were first incubated with rabbit anti-mouse Ikaros antibody

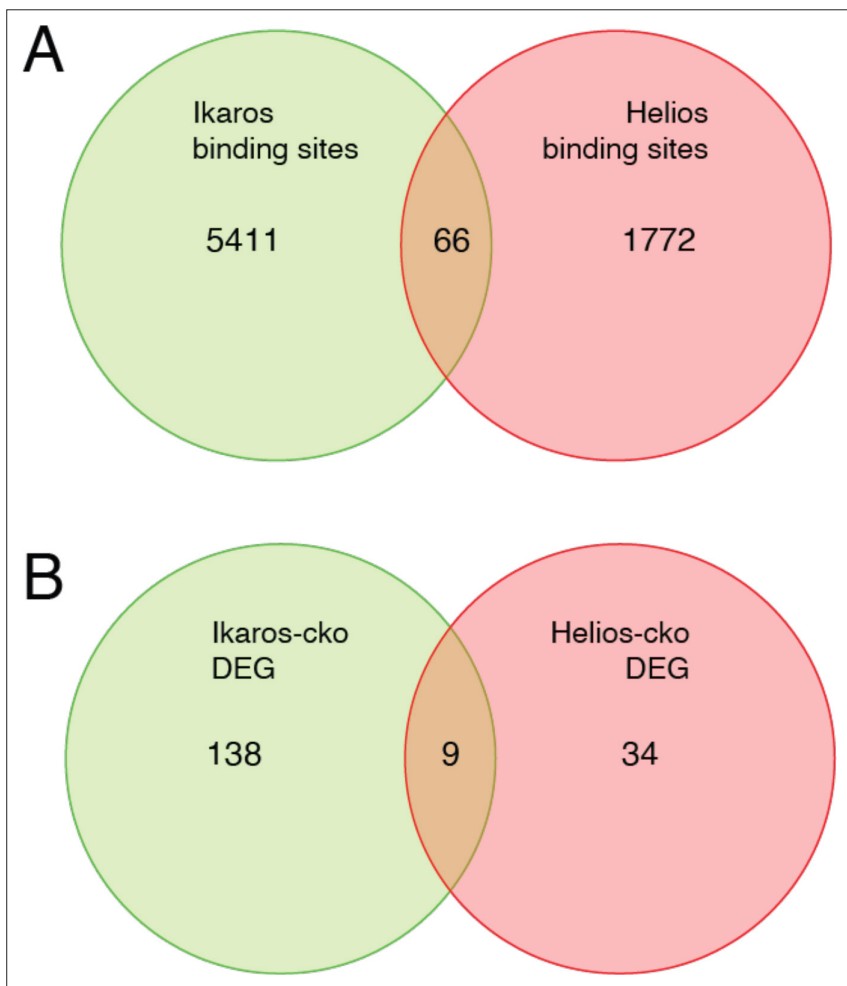

**Figure 10.** Comparison of Ikaros vs. Helios binding sites and regulated genes. Venn diagram in (**A**) depicts the comparison of published ChIP-seq Helios binding sites (*Kim et al., 2015*) (one replicate) with at least one bp of overlap with Ikaros binding sites from this study. Venn diagram in (**B**) depicts the overlap of differentially expressed genes (FDR <0.5 & abs(log2FC)>1) in Ikaros or Helios conditional knockout (*Yates et al., 2018*).

(Abcam) diluted 1:2000 in perm/wash buffer for 1 hr, and then cells were washed and stained with goat anti-rabbit-PE secondary antibody (1:2000 dilution) for 30 min. After washing, cells were stained for Foxp3, washed, and analyzed by flow cytometry on a Cytoflex equipped for multicolor detection. Flow cytometry data analysis was conducted with Flowjo10 software. Secreted IL-2 and IFNg in cell culture supernatants were determined by ELISA following the instructions provided by the vendor, Thermo Fisher Scientific.

### T cell and Treg purification

CD4+CD25-negative conventional T cells and CD4+CD25+Tregs were purified from spleen and lymph node cell single cell suspension using Miltenyi Treg and CD4 purification kits. For FACS sorting, total CD4+ T cells were isolated first using Miltenyi CD4+ T cell purification kit and then sorted on a FACS-Jaz sorter for CD4+YFP+ Tregs.

### Cell culture

Purified Treg were resuspended in RPMI 1640 medium supplemented with 10% FBS, 50 uM 2-ME, penicillin/streptomycin, and L-glutamine. Cells were stimulated with plate-bound mouse anti-CD3 and anti-CD28 (1 ug/ml each) in 96-well plates and incubated at 37°C in a cell culture incubator for the indicated times. For Treg proliferation assay, cells were labeled with Cell Trace (Thermo Fisher Scientific) and stimulated with microbeads coated anti-mouse CD3 /CD28 (Dynabeads mouse cell activator). Cell proliferation was determined after 3 days of activation. PKC inhibitor, Calphostin C (PKF) was purchased from Cayman Chemical Company. Tregs were cultured in the presence of various concentrations of PKF in 96-well plates coated with anti-CD3 and anti-CD28 (1 ug/ml each) for 3 days. Supernatant was collected for ELISA. For intracellular cytokine staining, cells were harvested and re-stimulated with PMA (15 ng), ionomycin (1 uM), and Golgistop for 5 hr. Cells were harvested, washed with 1X PBS, and then stained for live cells with Live-dead aqua stain followed by staining for flow cytometry.

### STAT5 phosphorylation assay

Splenocytes were isolated from WT and Ikzf1-fl-Foxp3-YFP-Cre mice. Cells were washed with 1X PBS and the pellet was resuspended in RPMI medium at $2 \times 10^6$ /mL. To induce STAT5 activation, aliquots of $10^6$ splenocytes were treated with recombinant mouse IL-2 (Sigma, cat # 11271164001) at 5–20 units/mL and cultured in 48 well plate for 30 min at 37°C. Stimulated and unstimulated cells were harvested and washed with 2 mL of FACS buffer and then cells were fixed with BD transcription factor fixation/perm buffer (cat #562574, BD Bioscience) for 20 min at room temperature. Cells were washed with 2 mL of FACS buffer and the cell pellet was fixed with 0.5 mL of 90% ice-cold methanol for 30 min on ice. Cells were spun down, removed methanol, and washed 2X with BD Perm/wash (1X) buffer. Cells were stained with an antibody cocktail prepared in 1 X BD Perm/wash buffer containing fluorochrome-conjugated antibodies against CD4, CD44, CD62L, CD25, Foxp3, and phospho-STAT5 (pY694). Cells were stained at room temperature for 45 min followed by washing with BD Perm/wash buffer. Cell pellet was resuspended in 350 ul wash buffer and analyzed by flow cytometry. pSTAT5 staining was analyzed on the gated Treg population.

### In vitro Treg suppression assay

Lymphocyte cell suspensions were prepared using the lymph nodes and spleen collected from the Foxp3-YFP-Cre, *Ikzf1*-fl-Foxp3-YFP-Cre, and C57BL/6 mice. Conventional CD4+CD25-negative and CD4+CD25+ Tregs cells were purified from the lymphocytes of wild-type and *Ikzf1*-cko mutant mice using Miltenyi Treg isolation kit (cat # 130-091-041). APCs were negatively selected from the lymphocytes of C57BL/6 mouse using Miltenyi CD90 (Thy1.2) cat #130-049-101 kit. APCs were gamma irradiated in a cesium irradiator. Ten million CD4+ Tconv cells were labeled with CellTrace Violet and resuspended in RPMI medium at $1 \times 10^6$ cells/mL. Labeled Tconv cells (50,000/well) were cocultured with $0.1 \times 10^6$/well-irradiated APCs plus various Treg:Tconv ratios (1:1, 1:2, 1:4, 1:8, 1:0) in 96-well round bottom plates. Cells were stimulated with soluble anti-CD3 (1 ug/mL) and cells were cultured at 37°C for 72 hr in a cell culture incubator. Cells were harvested, washed with 1 X PBS, and stained with live/dead aqua dye followed by flow staining with CD4, CD25, and CD44 fluorochrome conjugated antibodies. Cells were analyzed on a Cytoflex flow cytometer and data was analyzed by Flowjo10

software. Cell division was quantified as described previously (*Wells et al., 1997*), and percent suppression represents the reduction in cell division measured in the Tconv in the presence of Treg compared to no Treg.

## Co-immunoprecipitation analysis

HEK293T cells were co-transfected with eukaryotic expression vectors encoding Flag-Ik1, Flag-Ik7, Foxp3, or control empty vector. A Flag-Runx1 construct was co-transfected with Foxp3 as a positive control for co-precipitation. After 48 hr of transfection, a cell lysate was prepared and Flag antibody immunoprecipitation was done for the lysates using a Flag IP kit (Zigma). Pulldown products were immunoblotted for Flag protein and Foxp3.

## T cell transduction

T cell transductions were performed as described previously (*Chen et al., 2006*). Briefly, mouse CD4 + T cells were transduced with an empty vector, Foxp3 vector, or co-transduced with a retroviral vector expressing the dominant negative Ik7 isoform. After 3 days of transduction, cells were harvested and re-stimulated with plate-bound anti-CD3 and anti-CD28. Supernatant was collected for IL-2 and IFNg ELISA and cells were harvested for Foxp3 chromatin immunoprecipitation.

## ChIP-seq library generation and analysis

For transcription factor and H3K27ac ChIP-seq analysis, we used in vivo expanded Tregs generated in mice using IL-2/anti-IL-2 complexes. Anti-mouse IL-2 antibody (BE0043) was purchased from Bioxcell and recombinant mouse IL-2 (carrier-free, cat # 575408 from Biolegend). IL-2/anti-IL-2 complexes were prepared by mixing both reagents, incubating at 37 C for 30 min, and diluted with 1 X PBS. Foxp3-YFP-Cre and *Ikzf1*–fl-Foxp3-YFP-Cre mice (5–6 weeks old) were injected intraperitoneally with 200 ul of complex containing 2 ug IL-2 and 10 ug anti-IL-2. Each mouse received an injection daily for 3 days and was treatment-free for another 3 days before harvesting lymph nodes and spleens. For Ikaros ChIP-seq, total CD4+ T cells were negatively enriched using Miltenyi CD4 microbeads and then sorted for YFP+ Tregs by a FACS-Jaz sorter. Tregs purified through Miltenyi Treg purification kit were used for Foxp3, Ikaros, and H3K27ac-ChIP-seq. Naïve CD4+ T cells from Foxp3-YFP-Cre mice were also purified for Ikaros and H3K27ac ChIP-seq using a mouse CD4+ naïve purification kit purchased from Mitenyi Biotech. For all ChIP-seq experiments, three biological replicates of cells isolated from three individual mice were used. Chromatin immunoprecipitations were performed using ChIP-IT high-sensitivity kits (cat #53040, Active Motif) following the manufacturer's instructions. Briefly, $5 \times 10^6$ purified cells were fixed in a medium for 15 min at room temperature using complete cell fixative solution prepared using formaldehyde and cell fixative solution from the kit. The reaction was stopped by adding 1/20 media volume of stop solution, cells were washed with ice-cold PBS and a cell pellet was stored at –80°C for later use or cells were lysed in chromatin preparation buffer cells as described in the protocol. The cell pellet was resuspended in ChIP buffer and chromatin was sonicated by a QSonica Q800R sonicator with settings: amplitude 20%, pulse for 30 s on, 30 s off, for a total of 30 cycles. For input DNA preparation, 25 ul of the sonicated sample was removed and DNA was isolated as suggested in the protocol. An agarose (1%) gel electrophoresis was performed for DNA isolated from the input fraction to determine the sonication efficiency. ChIP-validated anti-mouse Ikaros antibody (cat #39355) and H3K27Ac antibody (cat #39133) were purchased from Active Motif. For Foxp3 ChIP-seq, eBioscience anti-mouse monoclonal antibody (cat#14-5773-82) was purchased from Thermo Fisher Scientific. The volume of the sheared chromatin was adjusted to 200 ul using ChIP buffer, and to which was added 5 ul of Protease inhibitor, and a mix containing ChIP antibody (4 ug) and 5 ul of blocker, mixed and pre-incubated at room temperature for a minute. Final volume of the ChIP reaction was 240 ul, which was incubated at 4°C overnight on a rotator. Antibody-precipitated chromatin immune complexes were collected using washed protein G agarose beads and immune complexes were washed 5X in ChIP filtration columns using wash buffer, and eluted the DNA with elution buffer. The eluted DNA was reverse cross-linked and further purified through DNA purification columns. ChIP'd DNA was eluted from the column using 30 ul of DNA purification elution buffer. All ChIP-seq and input DNA libraries were made using a ThruPLEX DNA-Seq kit (cat #R400674, Takara Bio, USA) following the manufacturer's instructions. In brief, the fragmented DNA obtained from ChIP reaction or input DNA was end-repaired to generate blunt ends, to which stem-loop adaptors

with blocked 5' ends are ligated. Libraries were amplified through high-fidelity amplification buffer mix and Takara dual indexing primers (cat# R400407). Finally, the amplified dual-indexed libraries were purified using AMpure XP beads (Beckman Coulter, Cat # A63880) at 1:1 ratio. The purified DNA was recovered from the beads using 20 ul of TE buffer. The library quality was checked on a bioanalyzer using a high sensitivity DNA Chip. Library DNA concentration was determined using Qubit. Dual-indexed ChIP-seq libraries were pooled and sequenced on the Illumina NovaSeq 6000 platform. Reads were aligned to mm9 using bowtie2 and duplicated reads were marked using Picard with parameters VALIDATION_STRINGENCY = LENIENT and ASSUME_SORTED = true and removed samtools. Library quality was accessed using samtools flagstat to assess library complexity and strand cross correlation to assess (*Kharchenko et al., 2008*). Peaks were called using MACS2 with the parameters -g mm9 –nomodel –p 0.01 –keep-dup_all with the – extsize estimated fragment size from the strand cross-correlation for each replicate of H3K27ac, Ikaros, or Foxp3 with matching input sample. Reads were subsequently filtered by the ENCODE mm9 blacklist regions. Within the condition peaks were filtered to ones found in at least two replicates. Binary comparison between different ChIP peaks and ATAC-seq was performed using the R package GenomicRanges (1.46.1) findOverlaps function. For differential comparisons of H3K27ac ChIP signal, reads were normalized against background (10 K bins) using csaw (v1.28, http://bioconductor.org/packages/release/bioc/html/csaw.html). Peaks with a cpm value less than 3.0 were removed from further differential analysis. Differential analysis was performed using glmQLFit approach in edgeR (v3.36.0) with the normalization scaling factors calculated from csaw. FDR <0.05 was used as the cutoff for statistical significance. Signal reproducibility between replicate samples was accessed using pairwise Pearson correlation tests and principal component analysis. Significant OCR overlapping with H3K27ac peaks were annotated as enhancers. Correlations between enhancer accessibility, H3K27ac ChIP signal, and expression of nearest gene were computed using Pearson correlation coefficient implemented in the R function cor.test. Super-enhancers were called using the rank ordering of super-enhancer (ROSE) algorithm (*Whyte et al., 2013*). Briefly OCR called by ATAC-seq were used as input regions and clustered by genomic coordinates with a 12.5 kb stitching window. Merged replicates of WT and Ikzf1-cko H3K27ac signal and input were used as a measure of enhancer activity. The signal is represented as input subtracted reads per million per basepair and then are ranked-ordered. The position in the ranked list where the change in signal (slope when x-axis is the super-enhancer rank and y- axis is signal) equals 1 is used to define super-enhancers by the rapid increase in enhancer activity. Super-enhancers were defined independently for WT and Ikzf1-cko H3K27ac data. ChIP peaks were annotated to their nearest gene based on linear genomic distance.

## RNA-seq library generation and analysis

Total CD4+ T cells isolated from individual WT YFP Cre+ and Ikzf1-Treg-cko mice (three biological replicates) were sorted for YFP+ Tregs on a FACS-Jaz sorter. For stimulation, Tregs were activated with plate-bound anti-CD3 and anti-CD28 (1 ug/ml each) for 4 hrs. Total RNA was isolated from the unstimulated and stimulated Tregs using Direct-zol RNA micro prep kit (Zymo Research). Quality of the DNase-treated total RNA was checked on a bioanalyzer. Ribosomal RNA was depleted from the total RNA using QIAseq fast select multi-RNA removal kit for mouse RNA (Qiagen) and then RNAseq libraries were made using NEB Next Ultra II Directional RNA library prep kit for Illumina. RNAseq library quality was checked on a high sensitivity bioanalyzer and dual indexed libraries were sequenced to 51 bp reads on the Illumina NovaSeq 6000 platform. The pair-end fastq files were mapped to genome assembly mm9 by STAR (v2.6.0c) (*Dobin et al., 2013*) for each replicate. Ensembl v67 mm9 annotation was used for gene feature annotation and the unadjusted read count for gene feature was calculated by htseq-count (v0.6.1) (*Anders et al., 2015*) with parameter settings -f bam -r pos -s reverse -t exon -m union. The gene features annotated as rRNAs were removed from the final sample-by-gene read count matrix. The differential analysis was performed in R (v3.3.2) using the edgeR package (v3.16.5) (*Robinson et al., 2010*). Briefly, the raw reads on gene features with total CPM (read counts per million) value of less than 3.66 (the bottom 25% gene features when comparing the highest count per condition across all samples) were removed from differential analysis. The trimmed mean of M-values (TMM) method were used to calculate normalization scaling factors and quasi-likelihood negative binomial generalized log-linear (glmQLFit) approach was applied to the count data and through pairwise comparisons of stimulated and unstimulated IK cko and WT. The

differential expression genes (DEGs) between were identified with cut-off FDR<0.05 and absolute logFC>1. TPM values were calculated for differentially expressed genes and scaled expression values (across rows) were depicted using the R package ComplexHeatmap (2.10.0). Immunologic signature gene sets annotated in MSigDB (v7.0) were used for gene set enrichment analyses. Statistical significance of gene set enrichment for up and down regulated were determined using the hypergeometric test (one-tailed), implemented in the R phyper function.

## ATAC-seq library generation and analysis

YFP + Tregs were FACS sorted from Foxp3-YFP-Cre and *Ikzf1*-fl-Foxp3-YFP-Cre mice. Half were stimulated with plate-bound anti-CD3 +anti-CD28 for 4 hr and half were left unstimulated. One hundred thousand cells were lysed with 50 ul of cold lysis buffer (10 mM Tris pH 7.4, 10 mM NaCl, 3 mM MgCl2, 0.1% IGEPAL CA-630) and centrifuged at 550 g for 10 min at 4 C. Supernatants were discarded and nuclear pellets were subjected to Tn5 transposition using Nextera DNA preparation kits (cat #1502812). DNA was purified from transposition reaction using Qiagen Min-Elute PCR purification kits. Transposed DNA fragments were PCR amplified and indexed using Nextera index kits. PCR reaction products were size selected using AMpure-XP beads and DNA fragments were re-suspended in 10 mM Tris-HCl. DNA concentration was determined by Qubit, and ATAC seq library quality was checked on a bioanalyzer. Dual-indexed libraries were sequenced on the Illumina NovaSeq 6000 platform. Open chromatin peaks were called using the ENCODE ATAC-seq pipeline (https://www.enco-deproject.org/atac-seq/). Briefly, pair-end reads from three biological replicates for each cell type were aligned to hg19 genome using bowtie2, and duplicate reads were removed from the alignment. Narrow peaks were called independently for each replicate using macs2 (-p 0.01 `--nomodel` `--shift` –75 `--extsize` 150 -B `--SPMR` `--keep-dup` all `--call-summits`) and ENCODE blacklist regions (ENCSR636HFF) were removed from peaks in individual replicates. Peaks from all replicates were merged by bedtools (v2.25.0) within each cell type and the merged peaks present in less than two biological replicates were removed from further analysis. Finally, ATAC-seq peaks from both cell types were merged to obtain reference open chromatin regions. Quantitative comparisons of wild-type and *Ikzf1*-cko open chromatin landscapes were performed by evaluating read count differences against the reference OCR set. De-duplicated read counts for OCR were calculated for each library and normalized against background (10 K bins of genome) using the R package csaw (v1.8.1). OCR peaks with less than 3.6 CPM support in the top 25% of samples were removed from further differential analysis. Differential analysis was performed independently using edgeR (v3.16.5). Differential OCR between cell types were called if FDR <0.05 and absolute log2 fold change >1.

## Transcription factor motif enrichment

Enrichment of known transcription factors binding motifs was determined for the differential sets of OCRs using the R package PWMEnrich (v4.30.0). Enrichment of differential OCRs were calculated using all OCRs in the background model. Sequences were extracted from the bioconductor genome reference BSgenome.Mmusculus.UCSC.mm9.masked (v1.3.99) using the R package Biostrings (v2.62.0). We used JASPAR2020 position-weight matrix database as the motif reference (*Fornes et al., 2020*). P values were adjusted using FDR. Transcription factor footing improves the confidence of TF binding over pure sequencing matching. We identified putative Ikaros and Foxp3 footprint using HINT-ATAC (Hmm-based IdeNtification of Transcription factor footprints). HINT-ATAC corrects for Tn5 cleavage bias using a HMM based-approach to identify de novo TF footprints. Replicate deduplicated ATAC-seq bam files were merged and used as input to identify TF footprints located in the consensus set of OCRs. The de novo TF footprints were then matched to known TF PWMs in the JASPAR2020 database.

## DNA methylation analysis

To assess the natural thymic or peripheral origin of Treg, Foxp3 TSDR region which is fully demethylated in natural Tregs, was analyzed for CpG methylation by sodium bisulfite sequencing method. DNA methylation level at IFNg intronic enhancer was also analyzed. Briefly, 1 ug of DNA extracted from FACS sorted CD4+YFP+ Treg, or CD4 conventional T cells was bisulfite converted following *Thomas et al., 2005*. The converted DNA was desalted using the Wizard DNA clean-up system (Promega), desulphonated, neutralized, and precipitated with ethanol. The Foxp3 CNS2-TSDR, the IFNg

promoter, and the IFNg intronic enhancer regions were PCR amplified from bisulfite-converted DNA using nested PCR primers as described (*Floess et al., 2007*; *Northrop et al., 2006*). Gel-purified PCR products were cloned into PGEM-T easy vector, and plasmid DNA individual clones were sequenced using SP6 primers.

## Adoptive transfer colitis model

To assess the in vivo suppressive capacity of Ikaros deficient Treg, adoptive T cell transfer experiments were set up with Rag1 KO mice. CD4+CD25-negative conventional T cells and Treg were purified from the splenic and lymph node lymphocytes of male WT YFPCre +or Ikzf1-fl-Foxp3-Cre+ mice. Batches of Rag1 KO male mice consisting of 5 mice per group were received retro-orbital injection of 1 million WT CD4+ Tconv along with (0.25X106) WT or Ikaros-deficient Treg. Recipient mice were weighed three times a week and observed for various IBD induced-clinical symptoms. Mice were sacrificed after 40 days of T cell transfer, spleen, mesenteric lymph nodes, and colon were removed. Small pieces of tissue were cut from the lower part of colon and were fixed in formal fixative for histopathological analysis. Single cell suspension was made from spleen and mesenteric lymph nodes from individual mice, lymphocyte cell density was estimated using a hemocytometer. Lymphocytes were stained for CD4, CD8, CD25, CD44, and Foxp3 and analyzed by flow cytometry. Absolute T cell count was estimated using hemocytometer count and cell frequency derived from the flow cytometry analysis. Variation in weight curve between groups were statistically analyzed by ANOVA (GraphPad Prism).

## Cardiac transplantation

Fully MHC-mismatched BALB/c hearts (H-2d) were transplanted heterotopically into Foxp3-YFP-Cre and *Ikzf1*-fl-Foxp3-YFP-Cre mice by anastomosis of the donor ascending aorta and pulmonary artery to the recipient infrarenal aorta and pulmonary artery. On days 0, 2, and 4 of posttransplant, the transplant recipient mice were given i.p injection of CTLA4 Ig fusion protein (200 ug) and CD154 (200 ug). A separate batch of cardiac recipient mice were administered donor-specific transfusion (DST) with Balb/c splenocytes (5 million cells/recipient) and a single dose of MR1 (250 ug). Cardiac graft survival was determined by abdominal palpation, and cessation of cardiac contraction was considered as rejection of the graft. Grafts were monitored for 100 days, and graft survival data were analyzed by Kaplan-Meir/log-rank methods. For histopathological analysis, a separate batch of mice consisting of 3 mice per group were subjected to same transplant procedure for a short duration and then these recipient mice were sacrificed, grafts were removed before rejection, fixed in 1 X formal solution, and analyzed by standard histopathological methods.

## Cardiac histopathology and intragraft gene expression analysis

For histopathologic analysis of cardiac graft and colon from IBD experiments, portions of tissues were fixed in Shandon formal-Fixx solution (1X), Thermo Scientific. Tissues were processed and embedded in paraffin. Histologic sections were cut at 4 um thickness and stained with hematoxylin and eosin (H&E) stain and or with alcian blue. Immunohistochemical staining was performed for CD4 and Foxp3 by the CHOP Pathology cores. Blinded histopathological evaluations were performed by a histopathologist. For intragraft gene expression analysis and histopathological analysis, a separate batch of mice consisting of 3 mice per group were subjected to same transplant procedure with a duration of 19 days for CTLA4Ig and MRI-treated mice and a duration of 14 days for MRI and DST-treated mice, based on the weakness of transplant heart palpitation. These transplant recipient mice were sacrificed, graft was removed, a portion of the cardiac graft was frozen immediately in liquid nitrogen and a second portion of the graft tissue was fixed in 1X formal solution and analyzed by standard histopathological methods. Total RNA was extracted from the homogenized graft tissue using Trizol. One microgram of RNA was treated with DNase to avoid DNA contamination. It was then reverse transcribed using iScript cDNA synthesis kit (cat #170–8891) purchased from Bio-Rad. Gene expression analysis for the cardiac graft was performed by qRT PCR using Fast SYBR Green mastermix on a Applied Biosystems step one plus realtime PCR system. Following primer pairs were used for the qRT PCR amplification: *Foxp3*-Exp-F; AAAAGGAGAAGCTGGGAGCTATG, *Foxp3*-Exp-R; GTGGCTAC GATGCAGCAAGAG, *IFNg*-Exp-F; TTGCCAAGTTTGAGGTCAACAA, *IFNg*-Exp-R; GCTGGATTCCGG CAACAG, *TNFa*-F; CTGTAGCCCACGTCGTAGC, *TNFa*-R; TTGAGATCCATGCCGTTG, *IL6*-F; TGTT

CTCTGGGAAATCGTGGA, *IL6*-R; CTGCAAGTGCATCATCGTTGT, *CXCR3*-F; TACCTTGAGGTTAGTG AACGTCA, *CXCR3*-R; CGCTCTCGTTTTCCCCATAATC, *CXCL10*-F; TGCCGTCATTTTCTGCCTCA, *CXCL10*-R; GGACCGTCCTTGCGAGAG, *Bcl6*-F; GTGTCCCCCAGTTTGTGTCA, *Bcl6*-R; TGGAGCAT TCCGAGCAGAAG, *DGKA*-F; CAAACAGGGCCTGAGCTGTA, *DGKA*-R; CGAGACTTGGCATAGG TGCT, *CD27*-F; GGATGTGTGAGCCAGGTACA, *CD27*-R; GGGTGTGGTAGTCTGGAGAG, *m18S* RNA-F; TTCGAACGTCTGCCCTATCAA, *m18S* RNA-R; ACCCGTGGTCACCATGGTA.

## Acknowledgements

Funding was provided by NIH grants (AI059881, AI065881), the Fred & Susanne Biesecker Pediatric Liver Center, and the Center for Spatial and Functional Genomics at The Children's Hospital of Philadelphia. The authors have no competing interests to declare.

## Additional information

### Funding

| Funder | Grant reference number | Author |
|---|---|---|
| National Institutes of Health | AI065881 | Struan FA Grant |
| Children's Hospital of Philadelphia | | Andrew D Wells Struan FA Grant |
| National Institutes of Health | AI059881 | Andrew D Wells |

The funders had no role in study design, data collection and interpretation, or the decision to submit the work for publication.

### Author contributions

Rajan M Thomas, Conceptualization, Supervision, Visualization, Project administration, Writing – review and editing; Matthew C Pahl, Resources, Software, Supervision, Visualization, Project administration, Writing – review and editing; Liqing Wang, Visualization, Project administration; Struan FA Grant, Supervision, Funding acquisition, Project administration, Writing – review and editing; Wayne W Hancock, Supervision, Funding acquisition; Andrew D Wells, Conceptualization, Funding acquisition, Project administration, Resources, Supervision, Visualization, Writing – review and editing

### Author ORCIDs

Rajan M Thomas ⓘ http://orcid.org/0000-0003-2652-9992
Andrew D Wells ⓘ http://orcid.org/0000-0002-3630-2145

### Ethics

This study was performed in accordance with the recommendations in the Guide for the Care and Use of Laboratory Animals of the National Institutes of Health. All of the animals were handled according to approved institutional animal care and use committee (IACUC) protocol (#22-000594) of the Children's Hospital of Philadelphia.

Joint Public Review: https://doi.org/10.7554/eLife.91392.3.sa1
Author response https://doi.org/10.7554/eLife.91392.3.sa2

## Additional files

### Supplementary files

• Supplementary file 1. Genes differentially expressed (DEG) between Ikzf1-cko and wild-type regulatory T cells (Treg) in stimulated and unstimulated conditions measured by RNA-seq. Genes are annotated to Ensembl mm9/NCBIM37 release 67.

- Supplementary file 2. Differential expression gene (DEG) with known roles in regulatory T cells (Treg) function.
- Supplementary file 3. Gene set enrichment using MSigDB GO biological processes and MsigDB immune annotations for the upregulated and downregulated sets of genes. Enrichment was determined using one-sided hypergeometric tests with FDR adjustment.
- Supplementary file 4. Coordinates and statistics genomic regions with differential accessibility (DAR) in Ikzf1-cko vs. wild-type regulatory T cells (Treg) measured by ATAC-seq. Differentially accessible regions (DAR) were annotated to the nearest gene TSS.
- Supplementary file 5. Enrichment statistics for genomic regions with differential H3K27 acetylation in Ikzf1-cko vs. wild-type regulatory T cells (Treg) measured by ChIP-seq.
- Supplementary file 6. Coordinates of super-enhancer regions identified using the ROSE algorithm in Ikzf1-cko vs. wild-type regulatory T cells (Treg) as measured by H3K27ac ChIP-seq density. Super-enhancers were annotated to the nearest gene TSS.
- Supplementary file 7. Annotation of Ikaros ChIP-seq peaks. Peak calls of Ikaros ChIP peaks, whether the peak was found in Tconv vs. Tregs specifically or common to both. Nearest gene, distance to nearest gene. In_ocr indicates whether the Ikzf1 peak overlaps with an open chromatin region (OCR).
- Supplementary file 8. TF motif enrichment analysis at differentially accessible region (DAR). Enriched TF name, family, and the associated expression value (log2TPM) in the indicated condition are shown. The PWMEnrich p-values were adjusted using FDR.
- Supplementary file 9. Foxp3 binding sites called in Ikzf1-cko vs. wild-type regulatory T cells (Treg) as measured by ChIP-seq. The coordinates (mm9) for each peak are indicated as well as the nearest gene and associated distance. In_ocr indicates whether the Foxp3 peak overlaps with an open chromatin region (OCR).
- MDAR checklist

## Data availability

All data supporting the manuscript are contained in the supplementary files. Sequencing data have been deposited in GEO under accession codes GSE200176, GSE200177, GSE200178 and GSE200179.

The following datasets were generated:

| Author(s) | Year | Dataset title | Dataset URL | Database and Identifier |
|---|---|---|---|---|
| Thomas RM, Pahl MC, Wang L, Grant SFA, Hancock WW | 2024 | Foxp3 depends on Ikaros for control of regulatory T cell gene expression and function | https://www.ncbi.nlm.nih.gov/geo/query/acc.cgi?acc=GSE200179 | NCBI Gene Expression Omnibus, GSE200179 |
| Pahl MC, Thomas R, Wells AD | 2024 | Foxp3 depends on Ikaros for control of regulatory T cell gene expression and function [RNA-Seq] | https://www.ncbi.nlm.nih.gov/geo/query/acc.cgi?acc=GSE200178 | NCBI Gene Expression Omnibus, GSE200178 |
| Pahl MC, Thomas RM, Wells AD | 2024 | Foxp3 depends on Ikaros for control of regulatory T cell gene expression and function [ATAC-Seq] | https://www.ncbi.nlm.nih.gov/geo/query/acc.cgi?acc=GSE200176 | NCBI Gene Expression Omnibus, GSE200176 |
| Pahl MC, Thomas R, Wells AD | 2024 | Foxp3 depends on Ikaros for control of regulatory T cell gene expression and function [ChIP-Seq] | https://www.ncbi.nlm.nih.gov/geo/query/acc.cgi?acc=GSE200177 | NCBI Gene Expression Omnibus, GSE200177 |

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
