## [Editor Report · eLife assessment]

This comprehensive study provides **valuable** information on the cooperation of Ikaros with Foxp3 to establish and regulate a major portion of the epigenome and transcriptome of T-regulatory cells. While the data are **compelling**, the evidence that these features are solely intrinsic, independent of the micro-environment, could be strengthened.

---

## [Referee Report · Joint Public Review]

This study investigates the role of Ikaros, a zinc finger family transcription factor related to Helios and Eos, in T-regulatory (Treg) cell functionality in mice. Through genome-wide association studies and chromatin accessibility studies, the authors find that Ikaros shares similar binding sites to Foxp3. Ikaros cooperates with Foxp3 to establish a major portion of the Treg epigenome and transcriptome. Ikaros-deficient Treg exhibits Th1-like gene expression with abnormal expression of IL-2, IFNg, TNFa, and factors involved in Wnt and Notch signalling. Further, two models of inflammatory/ autoimmune diseases - Inflammatory Bowel Disease (IBD) and organ transplantation - are employed to examine the functional role of Ikaros in Treg-mediated immune suppression. The authors provide a detailed analysis of the epigenome and transcriptome of Ikaros-deficient Treg cells.

These studies establish Ikaros as a factor required in Treg for tolerance and the control of inflammatory immune responses. The data are of high quality. Overall, the study is well organized, and reports new data consolidating mechanistic aspects of Foxp3 mediated gene expression program in Treg cells.

Strengths:

The authors have performed biochemical studies focusing on mechanistic aspects of molecular functions of the Foxp3-mediated gene expression program and complemented these with functional experiments using two models of autoimmune diseases, thereby strengthening the study. The studies are comprehensive at both the cellular and molecular levels. The manuscript is well organized and presents a plethora of data regarding the transcriptomic landscape of these cells.

Weakness:

The findings of markedly increased percentages of activated conventional T cells (CD44hi), major increases in TFH cells, and elevated serum Ig levels indicate disrupted immune homeostasis even in the absence of overt autoimmune manifestations seen in histopathology. Thus, some of the observed genetic changes observed by the authors are likely Treg cell extrinsic. Further, clear conclusions from the genome-wide studies are lacking.

---

## [Author Response]

The following is the authors’ response to the original reviews.

**eLife assessment**
This comprehensive study provides valuable information on the cooperation of Ikaros with Foxp3 to establish and regulate a major portion of the epigenome and transcriptome of T-regulatory cells. However, the characterization is incomplete in that incontrovertible evidence that these are intrinsic features regulating biological function and not outcomes of the inflammatory micro-environment of the genetically manipulated mice is missing.
**Public Reviews:**
This study investigates the role of Ikaros, a zinc finger family transcription factor related to Helios and Eos, in T-regulatory (Treg) cell functionality in mice. Through genome-wide association studies and chromatin accessibility studies, the authors find that Ikaros shares similar binding sites to Foxp3. Ikaros cooperates with Foxp3 to establish a major portion of the Treg epigenome and transcriptome. Ikaros-deficient Treg exhibits Th1-like gene expression with abnormal expression of IL-2, IFNg, TNFa, and factors involved in Wnt and Notch signaling. Further, two models of inflammatory/ autoimmune diseases - Inflammatory Bowel Disease (IBD) and organ transplantation - are employed to examine the functional role of Ikaros in Treg-mediated immune suppression. The authors provide a detailed analysis of the epigenome and transcriptome of Ikaros-deficient Treg cells.These studies establish Ikaros as a factor required in Treg for tolerance and the control of inflammatory immune responses. The data are of high quality. Overall, the study is well organized, and reports new data consolidating mechanistic aspects of Foxp3 mediated gene expression program in Treg cells.Strengths:The authors have performed biochemical studies focusing on mechanistic aspects of molecular functions of the Foxp3-mediated gene expression program and complemented these with functional experiments using two models of autoimmune diseases, thereby strengthening the study. The studies are comprehensive at both the cellular and molecular levels. The manuscript is well organized and presents a plethora of data regarding the transcriptomic landscape of these cells.

Response: We thank the reviewers for their careful review and feedback on our manuscript. We appreciate that the reviewers and editors recognize the strength and comprehensive nature of our in vivo, cellular, biochemical, and genome-wide molecular studies, which are well-organized in the manuscript. The acknowledgment of the complementary functional experiments in two models of inflammatory disease is also encouraging.

Weakness:The authors claim that the mice have no pathologic signs of autoimmune disease even at a relatively old age, yet mice have an increased number of activated CD4+ T cells and T-follicular helper cells (even at the age of 6 weeks) as well as reduced naïve T-cells. Thus, immune homeostasis is perturbed in these mice even at a young age and the eXect of inflammatory microenvironments on cellular functions cannot be ruled out. Further, clear conclusions from the genome-wide studies are lacking.

Response: We agree with the reviewers' comment regarding the absence of overt autoimmune pathologies in Ikzf1-fl/fl-Foxp3-Cre+ mice, despite the increased frequency of activated CD4+ T cells, TFH cells, and apparent perturbation of lymphocyte homeostasis, even at a young age. It is noteworthy that while Ikaros is implicated in various autoimmune diseases, our specific mouse model in which Ikaros expression is lost only in Tregs, may not lead to a strong autoimmune phenotype in part due to the controlled environment of an extra-clean, pathogen-free animal facility. This aligns with a related study by Ana et al (2019, J. Immunol: doi:10.4049/jimmunol.1801270) in Ikzf1-fl/fl-dLck-Cre+ mice with loss of Ikaros expression in all mature CD4+ T cells, including Tregs, that exhibit no overt signs of overt autoimmune disease. Moreover, our transcriptomic studies reveal that increased expression of inflammatory genes in Ikzf1-deficient Treg is coupled with the simultaneous upregulation of genes with positive roles in Treg function. This balance suggests a compensatory mechanism within Ikaros-deficient Tregs that maintains their suppressive function until encountering an inflammatory immune challenge, which eventually leads to loss of Treg suppressive function in Treg-specific Ikaros-deficient mice. Our studies clearly show that Ikaros has cell-intrinsic effects in Treg that also lead to cell-extrinsic effects mediated by secreted factors that are likewise regulated by Ikaros. This can be said about the function of any transcription factor in any cell type. Our data clearly support the conclusion from the genome-wide studies that Ikaros plays a major role in establishing the active chromatin landscape, gene expression profile, and function of regulatory T cells in mice.

The following recommendations consolidate the views of the three reviewers of the manuscript.The experiments suggested and, in some instances, fresh analysis, are thought necessary, so that the evidence of Ikaros-Foxp3 interactions regulating T-regulatory cell biology is comprehensive and solid. We hope the comments are useful to strengthen the comprehensive analysis reported in this submission.The primary concern is that the indications of inflammation in the mice (see points 1 & 2 below) do not reflect in the experiments or consequent conclusions. The gap in the data should be addressed by testing these interactions in an appropriate context for which suggestions are included.Please note that the title of the manuscript may be modified to reflect the use of mice as the system of study for this work.(1) The evidence of inflammation (increased CD4 and T follicular cells) reported in the work requires new experiments to rigorously examine the relationship between Ikaros and Foxp3 to rule out the possible impact of the (inflammatory) microenvironment of the mice (Please see: Zemmour et al., Nat. Immunology 22, 607, 2021). Two possible experimental systems in mice are suggested.a) The use of heterozygous female mice, which should be phenotypically normal due to the presence of 50% normal Treg. Or,b) The generation of bone chimeras between wild-type and deficient mice using congenic markers.

Response: We agree that immune dysregulation that develops in the mice with age or during an inflammatory insult due to loss of Ikaros function in the Treg lineage is an important part of the phenotype of the animals. Our studies show that loss of Ikaros function in Treg influences the gene expression program such that Treg now produce inflammatory cytokines and ligands capable of engaging receptors expressed on Treg and other cells. This likely results in autocrine and paracrine signaling that induces further metabolic and gene expression differences not observed in wild-type mice. Indeed, we report in the manuscript that a sizable fraction of the differentially expressed genes do not appear to be direct Ikaros targets, but rather are downstream of Ikaros target genes such as Il2, Ifng, Notch, and Wnt. The mosaic experiments suggested will be a useful topic of future studies. Importantly, we argue that no gene expression study involving modulation of transcription factor activity in an organism- or cell-based system can be designed to measure only the direct effects of that transcription factor in a manner isolated from any indirect, downstream effects on the expression of other genes. We suggest that our current data remain highly valuable, as they reveal real and relevant biology in physiologic in vivo systems that do not depend upon the use of heterologous models. The fact that loss of Ikaros has an effect not only on its direct targets, but on gene programs driven in turn by the indirect effects of Ikaros-regulated factors, has been acknowledged in the manuscript.

(2) Figs. 7 and S5 show accumulation of CD4 cells (activated, memory, Tfh, Tfr) in LNs and spleens of the Ikaros KO over time. This is accompanied by elevated Igs but without overt autoimmune disease. KO Tregs had equivalent suppressive activity as WT Tregs against WT TeX in vitro. However, TeX from KO mice were resistant to the suppressive eXects of WT or KO Tregs. The authors interpret this as due to the increased percentage of memory cells within the KO TeXs, although they did not formally prove this point. Figs. 9 and S6 show that Ikaros KO mice are unable to be tolerized for cardiac allograft survival using two diXerent standard tolerogenic regiments. The rejecting allografts are accompanied by increased T-cell infiltration and upregulation of inflammatory genes. The authors suggest there is increased alloantibody, but alloantibody does not seem to have been measured.

Response: We are currently exploring in more detail the dysregulation of humoral immunity in the Ikzf1-deficient Treg model and plan to report these results in a future study.

(3) Linked to the above, a comparison of the chromatin occupancy of Ikaros in resting and activated Tregs would inform on whether and how Ikaros occupancy changes with the activation status of Tregs. Since the authors use in vitro stimulation for RNAseq and ATAC seq, ChIP seq analyses under these matching conditions will greatly add to the quality of the study. Since "Foxp3-dependent", ie. diXerential gene expression in the Foxp3GFPKO cells (PMID: 17220874) gene expression has been shown to be not entirely the same as Treg signature (i.e. gene expression or Tregs compared to Tnv), it will be worth correlating Ikaros, Foxp3 co-occupied genes and the corresponding fate of their expression with Foxp3-dependent and independent Treg signature gene sets.

Response: The prior study by Gavin et al. referred to above used duplicate samples instead of the standard three or more replicates required for a robust differential analysis of gene expression. The two samples in this study are variable, and no statistically significant differential gene expression was found between the experimental groups when we subjected these data to current analysis methods. For this reason, we have elected not to compare these prior data with our current data, which are robust, reproducible, and analyzed using current statistical methods. Furthermore, the mice used for the prior study develop a fatal inflammatory disease (scurfy) and therefore the Treg examined in this study would be subject to a much stronger extrinsic inflammatory environment than the Treg in our study, as our mice show no overt disease even with age.

Further, the consequence of the cooperation between the two transcription factors that can be inferred from the experiments in the study remains unclear. It is suggested that the authors could first consider the ChIP seq data from Foxp3, Ikaros co- and diXerentially occupied genes, and then correlate with the ATAC seq and gene expression data to comment on the consequence of this cooperation.

Response: We find that Ikaros binding at a given region has a strong effect on accessibility, as reported in the manuscript, but that Foxp3 occupancy has less consequence, consistent with a prior study suggesting that Foxp3 largely utilizes the open chromatin landscape already present in the conventional CD4 T cell lineage (PMID:23021222). Our data suggest that the dominant effect of Ikaros on Foxp3 is at the level of chromatin occupancy.

(4) In the comparative analyses of Ikaros and Foxp3 co-occupied regions and gene expression outcome, the authors mention "A total of 4423 Foxp3 binding sites were detected in the open chromatin landscape of wild-type Treg (Supplementary Table 9), and this ChIP-seq signal was enriched at accessible Foxp3 motifs." It is unclear whether the authors focused on the ATAC seq data and only examined the open chromatin regions for this analysis. In that case, it is unclear why. More so because the Ikaros footprint is more apparent in regions where accessibility is reduced upon deletion of Ikaros.

Response: Foxp3 has been shown to bind primarily at open chromatin shared between Tconv and Treg, unlike the pioneer activity of other Fox family members (PMID: 23021222, biorXiv https://www.biorxiv.org/content/10.1101/2023.10.06.561228v2.full.pdf). Consistent with this, we found the majority of peaks were in open chromatin. The motif analysis is quantitative, not binary, and takes into account Foxp3 binding sites at regions considered open in either condition, which is why we can see enrichment of Foxp3 motifs at sites going from more open to less open in the absence of Ikaros.

(5) Comments on figures:The authors use MFI repeatedly in many of the figures for quantitation of antigen expression. This is misleading as several of the target antigens are normally expressed on a subpopulation of cells, e.g., Eos. Percent positive and MFI would be more relevant. Cytokine production should be presented by intracellular staining (e.g., IL-2, IFNg) as Elisa data does not allow one to determine the percentage of abnormally producing cells.

Response: We show both ICS and ELISA in this paper, preferring ELISA because it is much more quantitative than ICS.

Suppl. Fig. 1c - the panels do not correspond precisely to the legend or the text. At least one panel is missing. In Supp fig 1c, the authors plotted eXector Tregs, which are by definition CD62LloCD44hi, but the Y axis says CD44hiCD62Lhi. Is this a typo? Also on page 4, describing this data the authors mentioned Tfr, but the data is not shown in the Supp fig 1c.

Response: We thank the reviewer for catching these mistakes. We have corrected the typo in the figure panel for Supplementary Figure 1c. Follicular Treg data are indeed presented in Figure 7h, not Supplementary Figure 1, and we have corrected the text.

Fig. 2, which lists the diXerent categories of diXerentially expressed genes, it will be helpful if the authors add two columns indicating fold change and FDR values.

Response: These values are included in Table S1

Fig. 3c, the resolution of the histograms in the inset should be enhanced.Fig. 3d, a histogram of representative CTV dilution plots, and an explanation of how the quantifications were done may be included.Fig. 3e - not well labeled. Are these fold changes? Enrichments? Number of gene elements within the GO term that are aXected? Something else?Fig. 3f - presented out of sequence. The data are a little hard to understand as the color scale is so subtle and the colors so close to one another that it is not entirely clear which gene expressions are increased vs decreased. Other than the simple statement that the Ikaros KO causes numerous changes, there does not seem to be a more consistent message from this data panel.Fig. 4a, in addition to the bar graphs, it will be better to show the plots in a histogram, gated on Foxp3+ Tregs in WT and KO groups, with representative MFI indicated on top. The resolution of the scatter plots in this figure, as well as some others throughout the manuscript, may be improved. Please increase the resolution wherever necessary.Fig. 4b should include representative plots for cytokine production gated in Tconv (CD4+Foxp3-) cells.Figs. 5a-h, S2-3a-d, and Suppl. Tables S4-8 show a comprehensive ATAC-seq and ChIP-seq analysis of genes and chromatin occupied or regulated by Ikaros, comparing Tconv vs Treg, stimulated vs naïve, and WT vs KO cells. It is a comprehensive tour-de-force analysis, again showing the major eXects of Ikaros on the entire Treg landscape of gene regulation.Fig. S5h-j should be explained or labeled in more detail. The fonts are too small to read, even at 200% magnification; and the cell and gene comparisons are not entirely clear.Supp. Fig. S3e is not referred to in the text.Fig. S4a is very diXicult to read; the font and plotted points are too small.

Response: We have improved the clarity of the figures where necessary. We also indicate in the figure legends that full gene lists are to be found in the supplementary tables.

Page 8, "Regions that exhibit reduced accessibility in Ikzf1 cko compared to wild-type Treg are enriched for the binding motif for Ikaros and the motif for TCF1 (Figure 5g).... ". Is this Fig. 5i or 5g?

Response: This statement is correct and is referring to data depicted in Figure 5g.

In Fig 6e, Flag-Ik7 is not visible in any of the inputs. The co-IP between Foxp3 and Runx1 (presumably a positive control) is not eXicient in this experimental condition. Co-IP experiments performed in primary cells upon retroviral transduction of the tagged proteins to confirm observations in cell lines are suggested.

Response: Runx1 is shown to co-precipitate with Foxp3 as expected, although the band is not intense, and the data depicted are representative of 3 experiments. Ik7 was included in this transient transfection experiment as a redundant control, and the referee is correct that Ik7 did not express well in this experiment and cannot be seen in this exposure. We showed these blots intact in the spirit of not digitally altering the data, and because the low Ik7 expression did not impact our ability to demonstrate specific co-precipitation of Foxp3 with full length Ikaros (Ik1). The images include nearly the entire mini-blots, and we have added molecular weight markers for clarity. As indicated in the legend, the cytokine and ChIP data in 6f are from a separate model of retrovirally Foxp3/Ik7transduced T cells that we and others have used in multiple prior studies (e.g. Thomas JI 2007, Thomas JI 2010). The interpretability of these experiments is not impacted by the transient transfection data from figure 6e. It should be noted that a prior study by Rudra et al. that is cited and referred to in the manuscript used a similar approach to also establish that Foxp3 and Ikaros form a complex in cells.

In Fig 6f, the authors state that Foxp3 overexpression in CD4 cells results in promoter occupancy of both IL2 and IFNg, however, data shows only IL2. Also in 6f, Foxp3 overexpression reduces IL2 and IFNg secretion, measured by ELISA, which is recovered by IkDN. However, the eXect of Foxp3 along with WT Ikaros (which should not modulate, and if anything, further repress IL2, IFNg production) is not shown.

Response: The reviewer is correct that ectopic expression of Ikaros leads to repression of cytokine gene expression, which we and others have shown in prior studies. Because the focus of this study was on loss of Ikaros function in Treg, we did not elect to overexpress full-length Ikaros. However, we completely agree that Ikaros GOF in Treg is an important topic for future studies.

Fig. 7e-g, how is %suppression calculated? Can representative CTV dilution plots for the suppression assays be shown?

Response: Cell division was quantified as described previously (see ref 50), and percent suppression represents the reduction in cell division measured by Tconv in the presence of Treg compared to in the absence of Treg. This has been clarified in the methods section.

In Fig 8 and the supplementary figures the representative colon pictures (Fig. S6a-c) do not show convincing diXerences in colon morphology even though all the other histology and clinical parameters are clear. Are the figures mislabeled?In Fig 8c-e and other histology figures scale bars should be shown.Fig. 8c-e, the Alcian blue staining among the groups appears similar; perhaps this is due to the low power magnification.

Response: We have edited this figure for clarity

Additional comments:Fig 10 is explained in the discussion section for the first time. The authors may want to consider including this when introducing Ikzf1 ChIPseq data for the first time in the study.

Response: The reviewer raises a valid point but we have elected to retain the current organizational structure of the manuscript.

A more complete characterization of the activated conventional cells including both CD4+ and CD8+ T cells for cytokine production during aging may be considered, as it is highly likely that abnormalities in cytokine production will be observed.

Response: We agree and are planning additional such experiments in future studies focusing on in vivo models of tolerance.

The failure of suppression of T cell proliferation which the authors claim is due to the presence of activated memory T cells can be better documented by using naive responder cells from the cKO mice.

Response: We agree and are planning additional such experiments in a future study focusing on further aspects of cellular immunobiology impacted by Ikaros, but we will give preference to in vivo models of tolerance in such studies.